# Spatiotemporal Learning on Cell-embedded Graphs

## Abstract

Data-driven simulation of physical systems has recently kindled significant attention, where many neural models have been developed. In particular, mesh-based graph neural networks (GNNs) have demonstrated significant potential in predicting spatiotemporal dynamics across arbitrary geometric domains. However, the existing node-edge message passing mechanism in GNNs limits the model's representation learning ability. In this paper, we proposed a cell-embedded GNN model (aka, CeGNN) to learn spatiotemporal dynamics with lifted performance. Specifically, we introduce a learnable cell attribution to the node-edge message passing process, which better captures the spatial dependency of regional features. Such a strategy essentially upgrades the local aggregation scheme from first order (e.g., from edge to node) to a higher order (e.g., from volume and edge to node), which takes advantage of volumetric information in message passing. Meanwhile, a novel feature-enhanced block is designed to further improve the performance of CeGNN and alleviate the over-smoothness problem. The extensive experiments on various PDE systems and one real-world dataset demonstrate that CeGNN achieves superior performance compared with other baseline models, significantly reducing the prediction errors on several PDE systems.

## 1 Introduction

Solving Partial Differential Equations (PDEs) is often essential for analyzing and modeling complex spatiotemporal dynamic processes across various scientific and engineering fields. For example, weather prediction (Scher, 2018; Schultz et al., 2021; Grover et al., 2015), ocean current motion prediction (Zheng et al., 2020), nonlinear engineering structure earthquake response prediction (Zhang et al., 2019), material mechanical properties simulation (Wang & Sun, 2018), etc. Traditionally, classical numerical methods (e.g., Finite Difference Method (FDM) (Godunov & Bohachevsky, 1959; Özişik et al., 2017), Finite Volume Method (FVM) (Eymard et al., 2000), and Finite Element Method (FEM) (Hughes, 2012)) are utilized to solve these PDEs, requiring substantial analytical or computational efforts. Although this problem has been simplified via discretizing the space, the issue of trade-off between cost and precision intensifies when dealing with varying domain geometries (e.g., different initial or boundary conditions or various input parameters), especially in real-world scenarios. In the last few decades, Deep Learning (DL) (Pinkus, 1999; Tolstikhin et al., 2021; Albawi et al., 2017; Koutnik et al., 2014; Sundermeyer et al., 2012) models have made great progress in approximating high-dimensional PDEs benefiting from existing rich labeled or unlabeled datasets. However, there are certain drawbacks in this simple approach of learning the non-linear mapping between inputs and outputs from data. For example, their performance is severely limited by the training datasets, the neural network lacks interpretability and generalizes poorly.

Embedding domain-specific expertise (e.g., Physics-informed Neural Networks (PINNs) (Raissi et al., 2019)) has shown the potential to tackle these problems (Krishnapriyan et al., 2021; Gao et al., 2021; He et al., 2023; Li et al., 2024b). However, the core part of PINNs, *Automatic Differentiation* (AD) approach, has two major drawbacks: (1) it is necessary to formulate explicit governing equations into the loss function, and (2) the parameters in high-dimensional feature spaces cannot be efficiently optimized when facing highly complex networks like graph networks. As shown in Figure 1**e**, there are no any predefined equations available to represent the evolution patterns of sea surface temperature at varying depths. Neural Operators, such as DeepONet (Lu et al., 2021) and Latent

Figure 1: Examples of datasets, including classic governing equations and more challenging real-world dataset. **a**, the 2D Burgers equation. **b**, the 2D Fitzhugh-Nagumo equation. **c**, the 2D Gray-Scott equation. **d**, the 3D Gray-Scott equation. **e**, the 2D Black-Sea dataset.

DeepONet (Kontolati et al., 2024), have emerged as another paradigm to learn these complex non-linear behaviors. The most well-known models, Fourier Neural Operator (FNO) (Li et al., 2021a) and its variants (Tran et al., 2021; Wen et al., 2022; Ashiqur Rahman et al., 2023; Li et al., 2024a), utilize neural networks to learn parameters in the Fourier space for fast and effective turbulence simulation. Likewise, they inevitably have the same shortcoming as those traditional methods: over-reliance on data and biased towards the grid domain.

The other representative models, Transformer models (Vaswani et al., 2017; Wu et al., 2024) and GNNs (Liu et al., 2020; Gao et al., 2022; McCardle, 2023; Horie & Mitsume, 2024), have demonstrated significant influence in predicting spatiotemporal dynamics across arbitrary geometric domains. In particular, mesh-based graph neural networks (GNNs) (Gilmer et al., 2017; 2020; Pfaff et al., 2021; Brandstetter et al., 2022) learn vastly different dynamics of various physical systems, ranging from structural mechanics and cloth to fluid simulations. However, the existing node-edge message passing mechanism in GNNs overestimates the primary role of "message" passing function on the neighbor "edges", limiting the model's representation learning ability. In general, this strategy leads to highly homogeneous node features after multiple rounds of message passing, making the features ineffective at representing distinct characteristics, namely, the over-smoothness problem.

To further address the above issues, we proposed an end-to-end graph-based framework, Cell-embedded Graph Neural Network (CeGNN), to model spatiotemporal dynamics across various domains with improved performance. Specifically, after detecting discontinuities in space, we introduce a learnable cell attribution to the node-edge message passing process, which better captures the spatial dependency of regional features. Such a strategy essentially upgrades the local aggregation scheme from the first order (e.g., from edge to node) to a higher order (e.g., from volume and edge to node), which takes advantage of volumetric information in message passing. Meanwhile, a novel feature-enhanced (FE) block is designed to further improve the performance of CeGNN and relieve the over-smoothness problem, via treating the latent features as basis functions and further processing these features on this concept. In detail, it regards the node latent feature $\mathbf{h}_i$ as basis and builds a higher-order tensor feature via an outer-product operation, e.g., $\mathbf{h}_i \otimes \mathbf{h}_i$. This process creates abundant second-order nonlinear terms to enrich the feature map. We then use a mask operation to randomly sample these terms, filtering the appropriate information by a learnable weight tensor to enhance the model's representation capacity. Figure 2 shows an outline of our proposed model. Our extensive experiments on many PDE-centric systems and real-world datasets show that CeGNN can significantly enhance spatiotemporal dynamic learning in various scenarios, particularly with limited datasets. The key contributions of this paper are summarized as follows:

- We introduce cell attributions legitimately to learn second-order information from connected nodes of any cell, allowing us to rapidly identify non-local relationships that traditional message-passing mechanisms often fail to capture directly.
- We propose the FE block to enrich the feature representations and filter more effective information via learnable parameters.
- Our approach stands out for its lower error, better interpretability, and robust generalizability, making a substantial progress in the spatiotemporal dynamic field.

## 2 RELATED WORKS

Spatiotemporal dynamics research, as one of the important frontier research areas, is integral to fields ranging from traditional fluid dynamics to economics and finance. In this part, we firstly give a brief introduction to spatiotemporal PDEs. Then, the relevant progress in spatiotemporal dynamics research is described from the perspectives of classical and neural solvers.

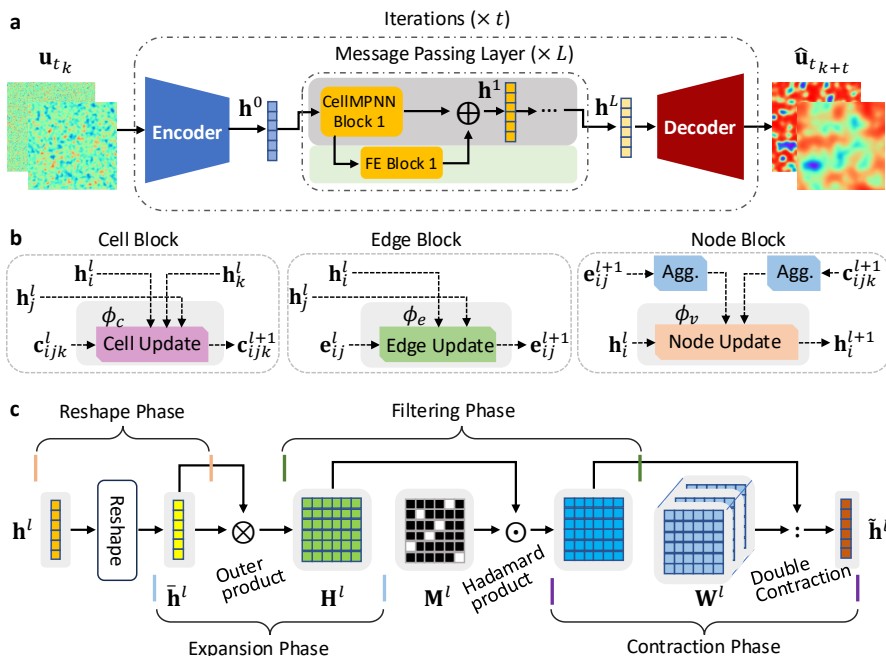

Figure 2: Network architecture of CeGNN. **a**, an encoder encodes the physical variables to latent features, a message passing block processes these latent features iteratively, and a decoder maps back to the physical states. **b**, three components in the CellMPNN block. **c**, the process of FE block.

## 2.1 PDE FORMULATION

Without loss of generality, the time-dependent PDEs generally describe the evolution of a continuous field over certain time intervals, which can be cast into the following form: $\partial \mathbf{u}/\partial t = \mathcal{F}\left(\theta, t, \mathbf{x}, \mathbf{u}, \nabla \mathbf{u}, \nabla^2 \mathbf{u}, \dots\right)$, where $\mathcal{F}(\cdot)$ denotes an unknown linear or nonlinear function comprised of the spatiotemporal variable $\mathbf{u}(\mathbf{x}, t) \in \mathbb{R}^d$, its corresponding partial derivatives (e.g., $\nabla \mathbf{u}, \nabla^2 \mathbf{u}$), and some related parameters $\theta$. Here, $\mathbf{x} \in \mathbb{R}^m$ denotes the $m$-dimensional spatial coordinate, $t \in \mathbb{R}^1$ the time, $\nabla$ the Nabla operator, $\nabla^2$ the Laplacian operator. The PDE is subjected to specific initial and boundary conditions.

## 2.2 CLASSICAL SOLVERS

To solve time-dependent PDEs, a common way is the method of lines (MOL). By discretizing in all but one dimension, it allows solutions to be computed via methods and software developed for the numerical integration of ordinary differential equations (ODEs) and differential-algebraic equations (DAEs). Meanwhile, the multigrid method (Wu et al., 2020) is another algorithm for solving PDEs via a hierarchy of discretizations. Other classical numerical methods (e.g., Finite Difference Method (FDM) (Godunov & Bohachevsky, 1959; Özişik et al., 2017), Finite Volume Method (FVM) (Eymard et al., 2000), and Finite Element Method (FEM) (Cao et al., 1999)) have also been utilized for practical applications (Reich, 2000; Hughes, 2012).

## 2.3 NEURAL SOLVERS

**PINN Methods.** Two main approaches, Physics-informed Neural networks (PINNs) (Raissi et al., 2019; Krishnapriyan et al., 2021; He et al., 2023) and Physics-informed Neural Operators (Li et al., 2021b; Hao et al., 2023; Kovachki et al., 2023), were developed to learn fluid and solid mechanics. With formulating the explicit governing equation as the loss function, PINNs constrain the latent feature spaces to a certain range, effectively learning from small data or even without any labeled data. Such a novel method immediately attracts the attention of many researchers and has been utilized in a wide range of applications governed by differential equations, such as heat transfer problems (Cai et al., 2021), power systems (Misyris et al., 2020), medical science (Sahli Costabal et al., 2020), and control of dynamical systems (Antonelo et al., 2024).

**Neural Operators.** Neural operators (Lu et al., 2021; Kontolati et al., 2024) combine various basis transforms (e.g., Fourier, multipole kernel, wavelet (Gupta et al., 2021)) with neural networks to accelerate PDE solvers in diverse applications. For example, Fourier Neural Operator (FNO) and its variants (Tran et al., 2021; Wen et al., 2022; Ashiqur Rahman et al., 2023; Li et al., 2024b) learn parameters in the Fourier domain for turbulence simulation. Especially, geo-FNO (Li et al., 2024a) maps the irregular domain into an uniform mesh with a specific geometric Fourier transform to fit irregular domains. However, they all follow the assumptions of periodicity and time-invariance property, making them fail in complex boundaries.

**Transformer Methods.** As another paradigm, Transformer (Vaswani et al., 2017) and its "x-former" family (Jiang et al., 2023; Wu et al., 2024) have also been utilized to solve complex PDEs. Since the attention mechanism results in higher complexity, many researchers are trying to alleviate this issue through various means. For example, linear attention mechanism (Li et al., 2023a; Hao et al., 2023) is a well-known method to address this limitation. Although the above methods alleviate the need for specific domain expertise, they all share the same limitations: instability in long-range prediction and weak generalization ability.

**Graph Methods.** Abundant works about Graph Neural Networks (GNNs) (Liu et al., 2020; Gao et al., 2021; 2022; Horie & Mitsume, 2024) and geometric learning (Bronstein et al., 2017; Hajij et al., 2020; 2022; Horie & Mitsume, 2024) attempt to utilize customized substructures to generalize message passing to more complex domains. For example, graph kernel methods (Anandkumar et al., 2020; Li et al., 2020) try to learn the implicit or explicit embedding in Reproducing Kernel Hilbert Spaces (RKHS) for identifying differential equations. (Belbute-Peres et al., 2020) considers solving the problem of predicting fluid flow using GNNs. Message Passing Simplicial Networks (MPSNs) (Bodnar et al., 2021) perform message passing on simplicial complexes (SCs). Most notably, Message Passing Neural Networks (MPNNs) (Gilmer et al., 2017; 2020; Janny et al., 2023; Perera & Agrawal, 2024) are utilized to tackle these issues, which learn latent representations on graphs via message passing mechanism. Especially, MeshGraphNets (Pfaff et al., 2021) and MP-PDE solver (Brandstetter et al., 2022) are two representative examples. While powerful and expressive, we still find that there are some problems with these methods: their expression ability is still not strong enough, their prediction error is still somewhat high, and they still rely on a large amount of data. Hence, we designed our model to address these three problems.

## 3 METHODOLOGY

In this section, we illustrate how our method effectively learn the solution of spatiotemporal PDEs under various parameters (e.g., the initial or boundary conditions, constant or variable coefficients) for a given physical system. All the source code and data would be posted after peer review.

### 3.1 NETWORK ARCHITECTURE

To enhance the performance of long-range prediction, we adapt the conventional "Encoder-Processor-Decoder" framework in (Pfaff et al., 2021; Brandstetter et al., 2022) as the backbone of our method, which is primarily designed to effectively learn the complex spatiotemporal dependencies on graphs. As shown in Figure 2, our proposed method mainly consists of the Feature-Enhanced (FE) block and the Cell-embedded MPNN (CellMPNN) block. These two key components update features in a sequential process to achieve the cascaded enhancement effect, described as follows: (1) Updating node-edge-cell features with the CellMPNN block; (2) Enriching higher-order node features with the FE block; (3) Iteratively repeating the above two steps until the specified number of processor layers is reached. The synergy of these two sequentially placed blocks in turn improves the model's representation learning capacity and generalization ability.

#### 3.1.1 FEATURE-ENHANCED BLOCK

For the sake of brevity and clarity, this block, shown in Figure 2, is designed to enhance the latent features from the upstream block and further alleviate the over-smoothness issue commonly seen in GNNs due to excessive aggregation. Please see the ablation results on FE's efficacy in Table 3. More details are provided in Appendix Section C.3.

**Outer Product as Basis Expansion.** The outer product operation $\otimes$ on the reshaped feature map $\overline{\mathbf{h}}_i \in \mathbb{R}^{D \times 1}$ expands the original latent feature space into a higher-order tensor space. This expansion

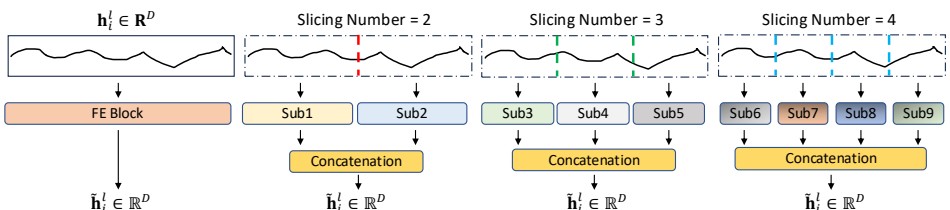

Figure 3: A scheme for reducing the number of parameters in FE block. Its quantitative experiment of the impact of window size and number of sub-features on CeGNN is shown in Table 5.

introduces second-order terms (e.g., $\alpha\beta$ for $\{\alpha, \beta\} \in \overline{\mathbf{h}}_i$), which can capture interactions between individual components of the original feature $\mathbf{h}_i \in \mathbb{R}^D$. Mathematically, the second-order tensor reads $\overline{\mathbf{h}}_i \otimes \overline{\mathbf{h}}_i$. This operation creates a richer feature map with cross-term interactions that may not be explicitly encoded in the original latent space.

**Lemma 1 (Nonlinear Representation)** *The second-order terms $\alpha\beta$ can model nonlinear dependencies between features. This is particularly useful for capturing complex interactions that linear transformations (e.g., via simple dot products) might overlook.*

**Definition 1** *The FE block expands the latent feature $\overline{\mathbf{h}}_i \in \mathbb{R}^{D \times 1}$ of node $i$ into a higher-order feature map $\mathbf{H}_i \in \mathbb{R}^{D \times D}$ using an outer product: $\mathbf{H}_i = \overline{\mathbf{h}}_i \otimes \overline{\mathbf{h}}_i$.*

**Regularization via Masking.** Masking introduces sparsity in $\mathbf{H}_i$, reducing overfitting. If $\mathbf{M}_{jk}$ is selected, $\mathbf{M}_{jk} = 1$. Otherwise, $\mathbf{M}_{jk} = 0$. Here $j$ and $k$ index the $M \in \mathbb{R}^{D \times D}$ components.

**Learnable Filtering.** The learnable weight tensor $\mathbf{W} \in \mathbb{R}^{D \times D \times D}$ acts as a filter, selecting and emphasizing the most informative terms.

**Definition 2** *A mask tensor $\mathbf{M} \in \mathbb{R}^{D \times D}$ is applied to randomly sample elements in $\mathbf{H}_i$, and the resulting masked tensor is processed using a learnable weight tensor $\mathbf{W} \in \mathbb{R}^{D \times D \times D}$ as follows: $\tilde{\mathbf{h}}_i = (\mathbf{M} \odot \mathbf{H}_i) : \mathbf{W}$, where $\odot$ represents element-wise multiplication and $:$ denotes double contraction of tensors. The resulting feature $\tilde{\mathbf{h}}_i \in \mathbb{R}^{1 \times D}$ enriches the representation of $\mathbf{h}_i \in \mathbb{R}^D$.*

**Corollary 1 (Representation Power)** *The full feature map $\mathbf{H}_i$ contains $D^2$ terms for a $D$-dimensional input feature vector $\mathbf{h}_i$. After masking, the effective representation space reduces by the sparsity of $\mathbf{M}$. The learnable filter $\mathbf{W}$ further narrows this down to the most critical terms.*

Considering the GPU memory requirements caused by additional parameters in the FE module, we further provide a feature splitting scheme within latent features to dramatically reduce the number of parameters and computation cost caused by the full FE block (see Figure 3). This strategy divides every feature into multiple sub-features with different window sizes, and then processes each part separately. Finally, these sub-features are combined for next layer learning. Note that we utilize the simplest window method to split the features in this article. Please see the ablation results on the effect of the window size and number of sub-feature on the feature splitting scheme in Table 5.

### 3.1.2 CELL-EMBEDDED GRAPH NEURAL NETWORK

Generally, the traditional message passing (MP) mechanism can be regarded as a refinement on a discrete space, analogous to an interpolation operation, which implies that edges are essentially interpolated from nodes. A MP mechanism introducing the cell has potential to further enhance the refinement of the discrete space (namely, secondary refinement), thereby reducing the magnitude of discretization errors spatially and paving the way for its application in complex graph structures.

**Definition 3 (Cell in Graph)** *Let $G = (V, E)$ be a graph, where $V$ is the set of nodes $\mathbf{v}$ and $E \subseteq V \times V$ is the set of edges. A cell in $G$ is a subset of nodes $C \subseteq V$, such that the nodes in $C$ form a complete subgraph (clique) or satisfy predefined structural relationships. In particular, a $k$-cell $C_k$ in a graph $G$ contains $k + 1$ nodes, where $\forall i, j \in C_k$, $(\mathbf{v}_i, \mathbf{v}_j) \in E$, representing various structures, such as node ($k = 0$), edge ($k = 1$), triangle ($k = 2$), tetrahedron ($k = 3$), and so on.*

**Corollary 2 (Expressive Power)** *Given a graph $G$ including many $k$-cell ($k = 0, 1, 2, \dots$), there exists a cell-based scheme that is more expressive than Weisfeiler-Lehman (WL) tests in distinguishing non-isomorphic graphs (see the proof in Supplementary C.2).*

Therefore, we proposed a new two-level cell-embedded mechanism to process the message on graphs. A new framework with cell-embedded features is designed as the following description.

**Encoder.** The encoder block maps the low-dimensional variables to corresponding high-dimensional latent features via differential functions (e.g., MLPs). The initial node feature $\mathbf{h}_i^0$ includes the node feature, one-hot feature of node type, and their position information. The initial edge feature $\mathbf{e}_{ij}^0$ contains the relative position vector, the distance of neighbor nodes, and etc. The initial cell feature $\mathbf{c}_{ijk}^0$ involves the centroid position of cell, the area of

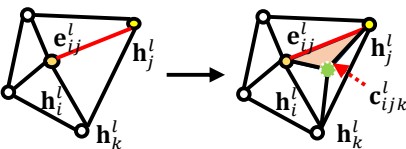

Figure 4: Cell on graphs. Green point represents the centroid of cell.

cell, and the relative position vector from three nodes to the corresponding centroid position. The corresponding forms are described as follows:

$$\mathbf{h}_i^0 = \phi_v^{en}\left(\mathbf{u}_i, \mathbf{x}_i, \kappa_i, \dots\right), \tag{1a}$$

$$\mathbf{e}_{ij}^0 = \phi_e^{en}\left((\mathbf{x}_j - \mathbf{x}_i), d_{ij}, \dots\right), \tag{1b}$$

$$\mathbf{c}_{ijk}^0 = \phi_c^{en}\left((\mathbf{x}_i - \mathbf{x}_{\triangle ijk}), (\mathbf{x}_j - \mathbf{x}_{\triangle ijk}), (\mathbf{x}_k - \mathbf{x}_{\triangle ijk}), \mathbf{x}_{\triangle ijk}, A_{\triangle ijk}, \dots\right), \tag{1c}$$

where the learnable functions $\phi_v^{en}(\cdot)$, $\phi_e^{en}(\cdot)$, and $\phi_c^{en}(\cdot)$ are applied to learn the latent features of node, edge, and cell; $(\mathbf{x}_j - \mathbf{x}_i)$ a relative position vector between the nodes $i$ and $j$; $d_{ij}$ the relative physical distance; $\kappa_i$ the type of node $i$; the $\mathbf{x}_{\triangle ijk}$ the centroid position of the cell $\triangle ijk$; the $A_{\triangle ijk}$ the area of the cell $\triangle ijk$. In addition, $(\cdot, \cdot)$ denotes the concatenation operation.

**Processor.** The processor iteratively processes the latent features from the upstream encoder via the cell-embedded MPNN block. As discussed in the above part, the key contribution of cell-embedded MPNN block lies in the introduction of the concept of cell. Figure 4 depicts the cell on graphs. Then, we divided the original edge channel $\mathbf{e}_{ij}^l$ into two parts: itself and its adjacent cells. With this simple process, the node can exchanges information with itself (nodal info), its immediate neighbor edges (derivative info), and its adjacent cells (integral info). In general, the cell features $\mathbf{c}_{ijk}^{l+1}$ and edge features $\mathbf{e}_{ij}^{l+1}$ firstly learn the effective information from adjacent nodes features, and then are aggregated to formulate the next node states $\mathbf{h}_i^{l+1}$. The procedure is described by the following forms:

$$\mathbf{c}_{ijk}^{l+1} = \phi_c^l\left(\mathbf{h}_i^l, \mathbf{h}_j^l, \mathbf{h}_k^l, \mathbf{c}_{ijk}^l\right), \tag{2a}$$

$$\mathbf{e}_{ij}^{l+1} = \phi_e^l\left(\mathbf{h}_i^l, \mathbf{h}_j^l, \mathbf{e}_{ij}^l\right), \tag{2b}$$

$$\mathbf{h}_i^{l+1} = \phi_v^l\Big(\underbrace{\mathbf{h}_i^l}_{\text{nodal info}}, \underbrace{\sum_{j \in \mathcal{N}_i} \mathbf{e}_{ij}^{l+1}}_{\text{derivative info}}, \underbrace{\sum_{jk \in \mathcal{N}_i} \mathbf{c}_{ijk}^{l+1}}_{\text{integral info}}\Big), \tag{2c}$$

where $j \in \mathcal{N}_i$ represents every neighbor edge $\mathbf{e}_{ij}$ at node $i$; $jk \in \mathcal{N}_i$ every neighbor cell $\triangle ijk$ at node $i$. $\phi_c^l, \phi_e^l, \phi_v^l$ are the differential functions of cell, edge, and node. Note that we have reformulated the message passing mechanism, where edge and cell features, without interaction, are used to simultaneously update the node features. See the test of computational cost and scalability in Table 4.

**Decoder.** The decoder maps latent features back to physical variables on graphs. With a skip connection, we acquire new states $\mathbf{u}_{t_{k+1}}$ by incremental learning, described as follows: $\hat{\mathbf{u}}_{i,t_{k+1}} = \phi_v^{de}\left(\mathbf{h}_i^L\right) + \mathbf{u}_{i,t_k}$, where $\phi_v^{de}(\cdot)$ is a differentiable function and $L$ the total number of layers.

## 4 EXPERIMENT

### 4.1 DATASETS AND BASELINES

To evaluate the performance of CeGNN, we experiment on the classic physical problems and more challenging real-world scenarios, including Burgers equation, Gray-Scott Reaction-Diffusion (GS

RD) equation, FitzHugh-Nagumo (FN) equation, and Black-Sea (BS) dataset. The first three datasets are generated by various governing equations on the grid domain and the final one on irregular meshes. Here, the initial input fields of three synthetic datasets are generated on Gaussian distribution with various random seeds and the node connectivity is obtained by the Delaunay algorithm. Please see Supplementary Table S2 for a detailed description of these datasets. More details about dataset generation can be found in Appendix Section D.1. We compare our model with the most popular graph-based neural network, such as MeshGraphNet (MGN) (Pfaff et al., 2021), Graph Attention Network (GAT) (Velickovic et al., 2017), Graph Attention Network Variant (GATv2) (Brody et al., 2022), the state-of-the-art models, message passing neural PDE solver (MP-PDE) (Brandstetter et al., 2022) , Fourier Neural Operator (FNO) (Li et al., 2021a), Factorized FNO (FFNO) (Tran et al., 2021), Geometry-informed FNO (Geo-FNO) (Li et al., 2023b; 2024a), Transolver (Wu et al., 2024). As shown in Supplementary Table S3, a comparative analysis of these baselines is discussed. Additional detailed information about these baselines is described in Appendix Section D.2.

## 4.2 EXPERIMENTAL SETUP

In our experiments, we mainly focus on predicting much longer time steps with lower error and attempt to achieve better generalization ability of various initial conditions (ICs) and boundary conditions (BCs). For fairness, we set the feature dimension to 128 and utilize the one-step training strategy (i.e., one-step forward, one-step backward) for all tasks. All experiments are run on one NVIDIA A100 GPU. The Adaptive Moment Estimation (Adam) optimizer is utilized for the model training. All models are trained by injecting random Gaussian noise of varying standard deviation to the input to improve stability during rollout and correct small errors. Meanwhile, a Root Mean Square Error (RMSE) loss function is utilized to optimize parameters $\theta$ in networks. Given the ground-truth values $Y \in \mathbb{R}^{N*d}$ and the predictions $\hat{Y} \in \mathbb{R}^{N*d}$ at time $t$, the loss function is defined as follows: $RMSE(Y, \hat{Y}) = \sqrt{\frac{1}{N*d} \sum_{i=i}^{N} \sum_{j=1}^{d} (y_{ij} - \hat{y}_{ij})^2}$, where $y_{ij} \in Y$ and $\hat{y}_{ij} \in \hat{Y}$. More details about the experimental setup are described in Appendix Section D.3. The experimental results and parametric studies of our model are given in the following parts.

## 4.3 RESULTS

We consider three different types of study cases: (1) the generalization test, (2) the feature-enhanced effect, and (3) an ablation study. All our experiments revolve around the following questions: ***Can our model generalize well? Can our model achieve lower error with small data?***

**Generalization test.** We varied the initial input field, randomly sampled from a Gaussian distribution with various means and standard deviations, in order to test the generalization ability. According to the results in Figures 5 and 6, we found that CeGNN generalizes to different ICs robustly on all datasets. It is evident that the performances of CeGNN and all baselines in the multi-step long-term prediction vary significantly. However, the experiment results of FNO show that it performs relatively poorly on all datasets except for the Burgers equation. FFNO outperforms FNO across all datasets. Unexpectedly, GeoFNO, which incorporates the IPHI technique, achieves the worst performance. Similarly, Transolver also underperforms. From the experimental results of these models, we can infer that these methods on small datasets, exhibit poor generalization ability, falling short compared to graph-based methods. All results in Table 1 demonstrate that all graph-based models have great generalization ability. GATv2, the advanced variant of GAT, underperforms GAT on Burgers, FN, 2D GS RD, and BS datasets, but outperforms GAT on 3D GS RD equation, yet both methods fall short of MGN. Surprisingly, the performance of MP-PDE is mediocre. Although MP-PDE is trained by the multi-step prediction strategy during the training stage, its results are only slightly better than MGN on the BS dataset. In contrast, our method performs robustly with much smaller errors in the multi-step prediction problem on all datasets.

**Feature-enhanced effect.** We investigate the effectiveness of the FE block on all graph-based network over all datasets. The results are reported in Table 2, showing that the feature-enhanced block somewhat changes the performance of these networks. We can directly see that this module has improved the performance of all baselines on real-world datasets. Specifically, it achieves the best and worst promotions in the cases of "MGN + FE" on the 3D GS RD equation and "GATv2 + FE" on the 2D Burgers equation. Intriguingly, after embedding the FE block in the processor block, the attention-based graph networks (e.g., GAT and GATv2) perform worse on the governing equation

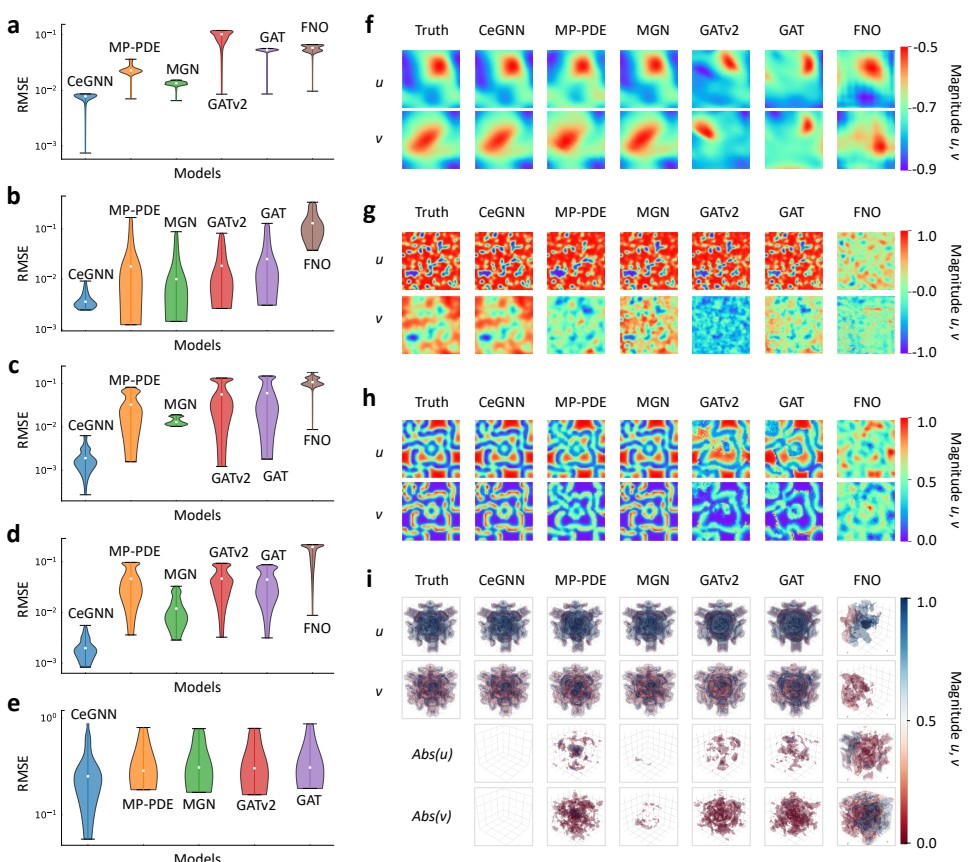

Figure 5: The test results of of all model on various datasets. **a-e**, the error distribution of test results. **f-i**, the slices of generalization test on four grid-based datasets. The results of the final irregular mesh-based dataset are displayed in Figure 6. The symbol $Abs(\cdot)$ represents a function for calculating the absolute error between ground-truth data and the prediction values.

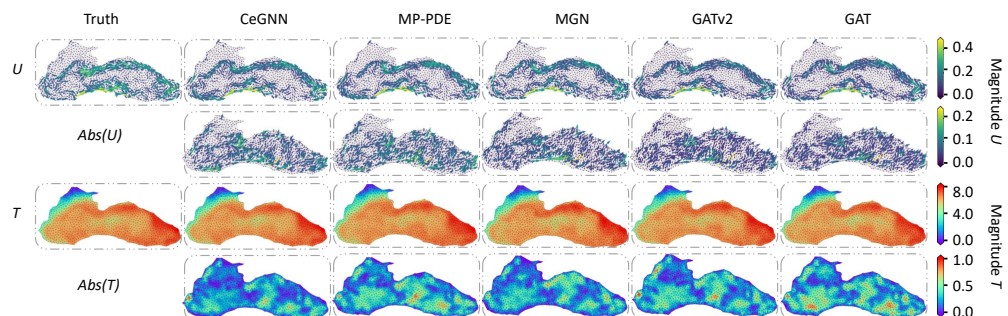

Figure 6: The snapshots of all models on BS Dataset, e.g., the rollout prediction at the 10th time step.

and better on the real-world dataset (even though the promotion is not large). This negative impact of the FE block on attention-based methods is essentially due to a logical conflict of design motivation. Consider two adjacent nodes $\mathbf{h}_1$ and $\mathbf{h}_2$ near a node $\mathbf{h}_0$, the attention mechanism assigns normalized weights $w_1$ and $w_2$ (e.g., $w_1 + w_2 = 1$) and aggregates features by the summation operation like $w_1\mathbf{h}_1 + w_2\mathbf{h}_2$. However, the FE block would disrupt this global normalization rule in the attention mechanism and reduces the expression capability of attention-based methods.

**Ablation study.** In this part, we perform an ablation study on all datasets to assess the contributions of the FE and CellMPNN blocks in CeGNN, as shown in Table 3. The results indicate that, without the introduction of the cell, the assembly of the tradition node-edge message passing mechanism

Table 1: Results of different methods. "–" represents that the model is unable or unsuitable to learn the dynamics directly. "↓" represents that the smaller the value of the quantitative metric, the better the model performance. The **bold** values and underlined values represent the optimal and sub-optimal results on various datasets. The promotion is calculated from the above two.

| Model | RMSE ↓ | | | | |
| --- | --- | --- | --- | --- | --- |
| | **2D Burgers** ($\Delta t = 0.001s$) | **2D FN** ($\Delta t = 0.002s$) | **2D GS RD** ($\Delta t = 0.25s$) | **3D GS RD** ($\Delta t = 0.25s$) | **2D BS** ($\Delta t = 1day$) |
| GAT | 0.11754 | 0.02589 | 0.07227 | 0.06396 | 0.62954 |
| GATv2 | 0.11944 | 0.03827 | 0.07301 | 0.04519 | 0.64796 |
| MGN | 0.01174 | 0.02108 | 0.02917 | 0.01925 | 0.61475 |
| MP-PDE | 0.01784 | 0.02848 | 0.03860 | 0.06528 | 0.60761 |
| FNO | 0.05754 | 0.12643 | 0.11331 | 0.17163 | – |
| FFNO | 0.03341 | 0.11921 | 0.03628 | 0.03594 | – |
| Geo-FNO (IPHI) | 0.59363 | 20.514 | 0.18669 | NaN | 1.2893 |
| Transolver | 0.17422 | 0.13724 | 0.18594 | 0.15204 | 0.81991 |
| Ours | **0.00664** | **0.00364** | **0.00248** | **0.00138** | **0.55599** |
| Promotion (%) ↑ | 43.4 | 82.9 | 91.4 | 92.8 | 8.4 |

Table 2: Efficacy of feature-enhanced block on the performance of graph-based baselines across all benchmarks with various methods.

| Model | RMSE ↓ | | | | |
| --- | --- | --- | --- | --- | --- |
| | **2D Burgers** ($\Delta t = 0.001s$) | **2D FN** ($\Delta t = 0.002s$) | **2D GS RD** ($\Delta t = 0.25s$) | **3D GS RD** ($\Delta t = 0.25s$) | **2D BS** ($\Delta t = 1day$) |
| GAT | 0.11754 | 0.02589 | 0.07227 | 0.06396 | 0.62954 |
| GAT + FE | 0.15132 | 0.02717 | 0.08527 | 0.07058 | 0.61984 |
| Promotion (%) ↑ | −28.7 | −4.9 | −17.9 | −10.3 | 1.5 |
| GATv2 | 0.11944 | 0.03827 | 0.07301 | 0.04519 | 0.64796 |
| GATv2 + FE | 0.18496 | 0.04117 | 0.09365 | 0.06432 | 0.63363 |
| Promotion (%) ↑ | −54.8 | −7.5 | −28.2 | −43.2 | 2.2 |
| MGN | 0.01174 | 0.02108 | 0.02917 | 0.01925 | 0.61475 |
| MGN + FE | 0.00817 | 0.01241 | 0.01583 | 0.00721 | 0.60593 |
| Promotion (%) ↑ | 30.4 | 41.1 | 45.7 | 62.5 | 1.4 |
| MP-PDE | 0.01784 | 0.02848 | 0.03860 | 0.06528 | 0.60761 |
| MP-PDE + FE | 0.01445 | 0.01957 | 0.02621 | 0.03655 | 0.60372 |
| Promotion (%) ↑ | 18.9 | 31.2 | 32.1 | 41.5 | 0.6 |

Table 3: Quantitative results of ablation study on CeGNN.

| Model | RMSE ↓ | | | | |
| --- | --- | --- | --- | --- | --- |
| | **2D Burgers** ($\Delta t = 0.001s$) | **2D FN** ($\Delta t = 0.002s$) | **2D GS RD** ($\Delta t = 0.25s$) | **3D GS RD** ($\Delta t = 0.25s$) | **2D BS** ($\Delta t = 1day$) |
| **w/o** Cell, FE | 0.01174 | 0.02108 | 0.02917 | 0.01925 | 0.61475 |
| **w/o** Cell | 0.00828 | 0.00982 | 0.01035 | 0.00680 | 0.58236 |
| **w/o** FE | 0.00877 | 0.00788 | 0.00803 | 0.00679 | 0.58271 |
| CeGNN (Full) | **0.00664** | **0.00364** | **0.00248** | **0.00138** | **0.55599** |

Table 4: Quantitative results of computational cost and scalability of CeGNN. Note that the symbol "Dim." means the the dimension of latent feature.

| Model | No.of layers | Dim. | Parameters | Training time (s/epoch) | GPU usage (GB) | 2D Burgers ↓ |
|---|---|---|---|---|---|---|
| CeGNN | 4 | 128 | 1,482,674 | 17.31 | 45.87 | **0.00664** |
| CeGNN w/o FE | 4 | 128 | 971,916 | 14.86 | 45.85 | 0.00877 |
| MGN | 4 | 128 | 514,236 | 8.64 | 32.56 | 0.01174 |
| MGN | 12 | 128 | 1,438,086 | 20.90 | 74.81 | 0.01858 |
| MP-PDE | 4 | 128 | 528,658 | 7.86 | 33.23 | 0.01784 |
| MP-PDE | 12 | 128 | 1,230,533 | 17.44 | 76.63 | 0.10101 |

Table 5: Quantitative results of the impact of window size and number of sub-features of CeGNN.

| Window size | Sub-features | Parameters | RMSE ↓ | | | |
|---|---|---|---|---|---|---|
| | | | 2D Burgers ($\Delta t = 0.001s$) | 2D FN ($\Delta t = 0.002s$) | 2D GS RD ($\Delta t = 0.25s$) | 3D GS RD ($\Delta t = 0.25s$) |
| 1 | 128 | 9,362,572 | 0.00665 | 0.00409 | 0.00251 | **0.00122** |
| 2 | 64 | 3,071,116 | 0.00714 | 0.00370 | **0.00232** | 0.00192 |
| 4 | 32 | 1,482,674 | **0.00664** | **0.00364** | 0.00248 | 0.00138 |
| 8 | 16 | 1,105,036 | 0.01263 | 0.02261 | 0.03481 | 0.04128 |
| 16 | 8 | 1,006,736 | 0.11753 | 0.12900 | 0.09120 | 0.18963 |

and FE block still shows good generalization ability, even with small data. Although we attempt to make the model learn higher-order information in each round, the results demonstrate that the cell-embedded mechanism did not achieve the desired performance, which means that the network architecture of traditional MPNNs still has the over-smoothness problems that need a better solution.

As shown in Table 4, our model with similar parameter ranges achieves the best performance compared with MGN and MP-PDE. We have verified that the improvement of our model performance is due to the innovative use of cell features rather than the introduction of additional parameters. In addition, as shown in Table 5, excessive feature segmentation can disrupt feature correlations and degrade model performance. Thus, we use the splitting method with a proper window size. We also have performed a data scaling test and report the results (data size vs. prediction error) in the Appendix Table S13 to support our claim of low data requirement. The tests were conducted on the Burgers example using 10, 20 and 30 trajectories as training data. It can be observed that our model with a smaller amount of data has equal or superior performance compared with that of other methods (MGN and MP-PDE) with larger amounts of data. Additionally, we have investigated the effectiveness of the cell feature on graph-based networks over all benchmarks. The results in Appendix Table S14 show the positive efficacy of cell features. A ablation test about the relative cell position information also demonstrates cell's significance, shown in Appendix Table S15. More importantly, the comparison results between two-level and three-level message passing mechanisms in Appendix Table S17 have verified the he rationality of model design motivation.

## 5 CONCLUSION

In this paper, we proposed an end-to-end graph-based framework (namely, CeGNN) to learn the complex spatiotemporal dynamics by utilizing the local information, CeGNN predicts future long-term unobserved states and addresses the over-smoothness problem in GNNs. Firstly, the learnable cell attribution in CellMPNN block captures the spatial dependency of regional features, upgrading the local aggregation scheme from the first order to a higher order. Secondly, the FE block enriches the node features, maintaining strong representational power even after multiple rounds of aggregation. The effectiveness of CeGNN has been proven through results on various datasets. Although CeGNN achieves superior performance on extensive experiments, there are several directions for future work, including that (1) pushing our model to learn on a finer mesh with more complex boundary conditions, and (2) further exploring the potential of cell attribution to learn higher-order information in a more refined way, rather than the rough handling in our article. We attempt to accomplish these goals in our future research work.

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

APPENDIX

## A  VARIABLES NOTATION

Moreover, we present a summary of the variable notations used in our paper, as detailed in Table S1.

Table S1: Variables notation used in our paper.

| Field | Variable Name | Short Name | Role |
|-------|--------------|------------|------|
| Global | x-component of velocity | $u(\mathbf{x}, t)$ | Input/Predicted |
| Global | y-component of velocity | $v(\mathbf{x}, t)$ | Input/Predicted |
| Global | vorticity | $w(\mathbf{x}, t)$ | Input/Predicted |
| Global | pressure | $p$ | Input/Predicted |
| Global | temperature in water depth of 12.5 meter | $T(\mathbf{x}, t)$ | Input/Predicted |
| Global | x-direction of space coordinate | $x$ | Input |
| Global | y-direction of space coordinate | $y$ | Input |
| Global | z-direction of space coordinate | $z$ | Input |
| Global | time coordinate | $t$ | Input |
| Global | time increment | $\Delta t$ | Input |
| Global | space increment | $\Delta x$ | Input |
| Global | discrete timestamp at $k$th step | $t_k$ | Input |

## B  BACKGROUND: OOD TEST IN SPATIOTEMPORAL PREDICTION

In this part, we would like to clarify the generalization tests with different random ICs in spatiotemporal dynamics are OOD. For a nonlinear PDE system, even if the ICs are IID, the spatiotemporal solution can be OOD, because of the following reasons.

**Nonlinearity and Emergence of Complex Patterns.** Nonlinear PDEs are characterized by their ability to produce highly complex behavior over time. This nonlinearity can amplify small differences in ICs, leading to the emergence of patterns or behaviors that are vastly different from what was initially expected. Even if the ICs are IID, the interactions dictated by the nonlinear terms in the PDE can cause the solution to evolve in a way that is not reflective of the initial distribution. As a result, the system may exhibit behaviors that are not represented in the original distribution, leading to an OOD trajectory dataset. For example, in the Burgers and FN examples, the ICs are generated based on Gaussian distribution with different random seeds (e.g., IID); however, the corresponding solution trajectories remain OOD judging from the histogram plots.

**Chaotic Dynamics.** Nonlinear PDEs may exhibit chaotic behavior. In these systems, small perturbations in ICs can lead to exponentially different solutions. Over time, this chaotic evolution can cause the solution to become highly sensitive to ICs (e.g., the 2D FN and 2D/3D GS RD test data examples shown in Figure 1b-d), resulting in a distribution that is very different from the IID distribution of ICs, yielding an OOD solution space.

**Long-term Evolution.** In many nonlinear PDEs, solutions tend to evolve toward certain stable structures or steady states known as attractors. These attractors can be complex structures in the solution space. Over time, the solution might converge to or oscillate around these attractors, regardless of the IID nature of ICs (e.g., the Burgers, FN and GS RD examples in our paper). The distribution of solutions near these attractors can be very different from the IC IID distribution. Essentially, the system's long-term behavior is determined more by the attractors rather than by ICs.

**Spatialtemporal Correlations.** The assumption of IID ICs implies no spatial/temporal correlations initially. However, the evolving dynamics governed by PDEs can introduce correlations over time. These correlations can lead to a solution that has a distribution quite different from the original IID distribution. The emergence of such correlations indicates that the evolved solution is not just a simple extension of ICs, producing datasets with OOD.

**Breaking of Statistical Assumptions.** As the system evolves, the assumptions that justified the IID nature of ICs may no longer hold. The dynamics of the PDE can induce structures, patterns,

or dependencies that were not present in ICs. As a result, the statistical properties of the evolved solution may diverge from those of ICs, leading to an OOD.

Hence, even though the ICs are IID, the resulting solution trajectories are OOD. This might be a little bit different from our understanding of IID/OOD datasets in common practices of NLP, CV, etc. In addition, we have indeed considered OOD ICs in our tests, e.g., the ICs of the 2D/3D GS RD examples are randomly placed square cube concentrations (1 or 2 square/cube blocks) as shown in Figure 1c-d. The resulting solution datasets are obviously OOD. Therefore, all the results of generalizing to different ICs represent OOD tests. Given the same comparison test sets, our model shows better generalization performance over other baselines.

## C  PROOFS

### C.1  SUPPLEMENTARY DEFINITIONS, LEMMAS, THEOREMS, COROLLARIES, OR PROOFS

**Lemma 2 (Feature Diversity)** *Introducing cells enhances feature diversity by encoding higher-order relationships among nodes. Specifically, the basic features in traditional MP mechanism are $\{\mathbf{h}_i, \mathbf{h}_j, \mathbf{e}_{ij}, \mathbf{e}_{ji}\}$, and the basic features in cell-based MP mechanism are $\{\mathbf{h}_i, \mathbf{h}_j, \mathbf{h}_k, \mathbf{e}_{ij}, \mathbf{e}_{jk}, \mathbf{e}_{ki}, \mathbf{c}_{ijk}, \mathbf{c}_{kij}, \mathbf{c}_{jki}\}$. The additional node $\mathbf{h}_k$ provides a richer context, enabling the capture of more complex patterns within one round.*

**Lemma 3 (Reduction in Ambiguity)** *Traditional MP methods rely solely on pairwise interactions, which can lead to ambiguity in cases of structural symmetry. Cell-based MP mechanism leverages the higher-order structure, reducing ambiguity by providing additional constraints through relationships between three or more nodes.*

**Corollary 3 (Improved Performance)** *Cell-based MP mechanism improves prediction capability by enhancing feature distinguishability through introducing higher-order relationships.*

**Corollary 4 (Suitability for Graphs)** *In dense or sparse graphs, cell-based methods outperform traditional methods by capturing multi-node interactions within one round, which are critical for preserving the graph's topology.*

**Explanation of Corollary 2.** 1-WL test relies on pairwise node comparisons and cannot distinguish graphs that are symmetric under pairwise relationships. By lifting graphs to a cell-embedded pattern and using cell-based MP mechanism, higher-order interactions are encoded, allowing discrimination of graphs that 1-WL test cannot separate. See the detail proof in Subsection C.2.

### C.2  WEISFEILER-LEHMAN (WL) TESTS

Weisfeiler-Lehman (WL) Test is an iterative graph isomorphism algorithm that updates node features by aggregating the features of neighboring nodes. After each iteration, the updated node features are hashed to encode structural information. Despite its effectiveness, WL test cannot distinguish certain non-isomorphic graphs, particularly when higher-order structural information is required. In this part, our goal is to show that cell-based message passing mechanism is more expressive than the 1-WL test for distinguishing non-isomorphic graphs.

**Task Definition.** A graph $G = (V, E)$ has a set of vertices $V$ and edges $E$. The 1-WL test iteratively computes node features $\mathbf{h}_i^l$ at iteration $l$ as $\mathbf{h}_i^{l+1} = \text{Hash}\left(\mathbf{h}_i^l, \{\mathbf{h}_j^l : j \in \mathcal{N}_i\}\right)$, where $\mathcal{N}_i$ is the set of neighbors of $i$, and $\mathbf{h}_i^0$ is initialized with the node's feature.

Consider two graphs $G_1$ and $G_2$ (see Figure S1), the WL test fails to distinguish $G_1$ and $G_2$ because it only aggregates local neighborhood information, and both graphs have identical degree distributions and neighborhood structures for all nodes.

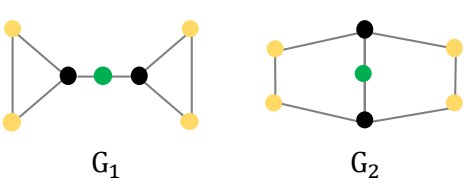

Given that cells (e.g., triangles) can be explicitly considered, we propose a simple cell-based scheme, described as follows.

Figure S1: Two undistinguished graphs by 1-WL test. Different colors represent different labels.

**Initialization.** Initializing node features $\mathbf{h}_i^0$ based on their labels or degrees and initializing higher-order cell features $\mathbf{c}_{ijk}^0$ by aggregating node features within the triangle.

**Message Passing.** Updating node and cell features iteratively. For example:

$$\mathbf{h}_i^{l+1} = \text{Hash}\left(\mathbf{h}_i^l, \{\mathbf{h}_j^l : j \in \mathcal{N}_i\}, \{\mathbf{c}_{ijk}^l : jk \in \mathcal{N}_i\}\right), \tag{S1a}$$

$$\mathbf{c}_{ijk}^l = \text{Aggregate}\left(\mathbf{h}_i^l, \mathbf{h}_j^l, \mathbf{h}_k^l\right), \quad \forall i,j,k \in \Delta_{ijk}. \tag{S1b}$$

**Expressiveness.** In $G_2$, no triangles exist, so all triangle-related features $\mathbf{c}_{ijk}$ will remain zero or absent. (2) In $G_1$, there are two triangles. These triangles generate non-zero features that propagate back to nodes during message passing. Thus, the presence of higher-order structures (triangles) allows the cell-based scheme to distinguish $G_1$ from $G_2$. And we proposed the following proposition about the expressiveness of cell-based message-passing scheme.

**Proposition 1** *The cell-based message-passing scheme is more expressive than 1-WL test because it captures higher-order interactions (e.g., triangles) that are invisible to 1-WL. This enhanced capability enables it to distinguish graphs, such as $G_1$ and $G_2$, which cannot be differentiated by 1-WL test.*

---

**Algorithm 1** Feature-enhanced block

---

**Input:** The current node states $\mathbf{h}^l \in \mathbb{R}^{N \times D}$
**Parameter:** The weight tensor $\mathbf{W}^l \in \mathbb{R}^{D \times D \times D}$, a mask matrix $\mathbf{M}^l \in \mathbb{R}^{D \times D}$
**Output:** The updated node states $\hat{\mathbf{h}}^l \in \mathbb{R}^{N \times D}$

1: **Reshape phase**. Reshape the input states $\mathbf{h}^l$ and obtain the $\overline{\mathbf{h}}^l \in \mathbb{R}^{N \times D \times 1}$.
2: **Expansion phase**. Play an Outer product operation on the $\overline{\mathbf{h}}^l$ to get new states $\mathbf{H}^l \in \mathbb{R}^{N \times D \times D}$.
3: **Filtering phase**. Filter feature with a Hadamard product operation on the new states $\mathbf{H}^l$ and the mask $\mathbf{M}^l$ to get new states $\hat{\mathbf{H}}^l \in \mathbb{R}^{N \times D \times D}$.
4: **Contraction phase**. Contract the new states $\hat{\mathbf{H}}^l$ with the weight $\mathbf{W}^l$ by a Double contraction operation to construct the final states $\tilde{\mathbf{h}}^l \in \mathbb{R}^{N \times D}$.
5: **return** The updated states $\tilde{\mathbf{h}}^l$.

---

### C.3 INSPIRATION OF FEATURE-ENHANCED (FE) BLOCK

The process of FE block is inspired by interaction models in physics and mathematically hypothesized to capture nonlinear dependencies, enhancing the model's representation power.

**Hypothesis.** The FE block's hypothesis could be framed as: By capturing second-order interactions between latent features and applying selective filtering, the model can better represent complex structures or relationships in the data.

**Physical Analogy.** In physics, the outer product and second-order terms are often used to model interactions, such as stress tensors in mechanics or pairwise correlations in quantum mechanics. Here, the module could draw an analogy to systems where interactions between individual components (features) are crucial to the overall behavior.

**Process Overview.** In detail, it regards the node latent feature $\overline{\mathbf{h}}_i \in \mathbb{R}^{D \times 1}$ as basis and builds a higher-order tensor feature $\mathbf{H}_i \in \mathbb{R}^{D \times D}$ via an outer-product operation, e.g., $\overline{\mathbf{h}}_i \otimes \overline{\mathbf{h}}_i$. This process in Algorithm 1 creates abundant second-order nonlinear terms to enrich the feature map. We then use a mask operation with $\mathbf{M} \in \mathbb{R}^{D \times D}$ to randomly sample these terms, filtering the appropriate information by a learnable weight tensor $\mathbf{W} \in \mathbb{R}^{D \times D \times D}$ to enhance the model's representation capacity.

## D SUPPLEMENTARY DETAILS OF EXPERIMENTS

### D.1 SUPPLEMENTARY DETAILS OF DATASETS

**Viscous Burgers Equation.** As a simple non-linear convection–diffusion PDE, Burgers equation generates fluid dynamics on various input parameters. For concreteness, within a given 2D field,

Table S2: Summary Information of Datasets. Note that the trajectory number for training, validation, and testing is described by the form like (5/2/3).

| Dataset | Space | PDE form | Boundary condition | Number of trajectories | Number of Nodes | Trajectory length | Force term |
|---------|-------|----------|--------------------|-----------------------|-----------------|-------------------|------------|
| Burgers | 2D | Eq. S2 | Periodic | 15 (10/2/3) | 2,500 ($50^2$) | 1000 | No |
| FN | 2D | Eq. S3 | Periodic | 10 (5/2/3) | 16,384 ($128^2$) | 3000 | Yes |
| GS RD | 2D | Eq. S4 | Periodic | 10 (5/2/3) | 2304 ($48^2$) | 3000 | No |
| GS RD | 3D | Eq. S4 | Periodic | 4 (1/1/2) | 13,824 ($24^3$) | 3000 | No |
| BS | 2D | – | N/A | 24 (20/2/2) | 1,000-20,000 | 365 | N/A |

there is a velocity $\mathbf{u} = [u, v]^T$ at per 2D grid point. Its general formulation is expressed as following description:

$$\frac{\partial \mathbf{u}}{\partial t} + \mathbf{u} \cdot \nabla \mathbf{u} = \mathbf{D}\nabla^2 \mathbf{u}, \tag{S2}$$

where viscosity $\mathbf{D} = [D_u, D_v]$ is the diffusion coefficient of fluid, $\mathbf{u}(\mathbf{x}, t)$ its velocity, $\mathbf{x}$ the 2D spatial coordinate and $t$ the 1D temporal coordinate. In this work, we generate the simulation trajectory within $\Omega \in [0, 1]^2$ and $t \in [0, 1](s)$, using a 4th-order Runge–Kutta time integration method (Ren et al., 2022) under the periodic condition. Here, we define $D_u = 0.01, D_v = 0.01, \Delta t = 0.001(s)$ and $\Delta x = 0.02$.

**Fitzhugh-Nagumo Equation.** Fitzhugh-Nagumo equation consists of a non-linear diffusion equation with different boundary conditions. We generates the training and testing data $\mathbf{u} = [u, v]^T$ in a given 2D field with periodic boundary conditions:

$$\frac{\partial u(\mathbf{x}, t)}{\partial t} = D_u \nabla^2 u + u - u^3 - v^3 + \alpha, \tag{S3a}$$

$$\frac{\partial v(\mathbf{x}, t)}{\partial t} = D_v \nabla^2 v + (u - v) \times \beta, \tag{S3b}$$

where $D_u, D_v$ are the diffusion coefficients, $\alpha, \beta$ the reaction coefficients. In this work, we generate the simulation trajectory within $\Omega \in [0, 128]^2$ and $t \in [0, 6](s)$, using a 4th-order Runge–Kutta time integration method (Ren et al., 2022) under the periodic condition. Here, we define $D_u = 1, D_v = 100, \alpha = 0.01, \beta = 0.25, \Delta t = 0.002(s)$ and $\Delta x = 1$.

**Gray-Scott Equation.** As a coupled reaction-diffusion PDE, Gray-Scott equation consists of a velocity $\mathbf{u} = [u, v]^T$. For example, given a 3D field, the corresponding form of each component on $\mathbf{x} = (x, y, z) \in \mathbb{R}^3$ and $t \in [0, T]$ is as follows:

$$\frac{\partial u(\mathbf{x}, t)}{\partial t} = D_u \nabla^2 u - uv^2 + \alpha(1 - u), \tag{S4a}$$

$$\frac{\partial v(\mathbf{x}, t)}{\partial t} = D_v \nabla^2 v + uv^2 - (\beta + \alpha)v, \tag{S4b}$$

where $D_u$ and $D_v$ are the variable diffusion coefficients, $\beta$ the conversion rate, $\alpha$ the in-flow rate of $u(\mathbf{x}, t)$ from the outside, and $(\alpha + \beta)$ the removal rate of $v(\mathbf{x}, t)$ from the reaction field. In this work, we generate the 2D simulation trajectory within $\Omega \in [0, 96]^2$ and the 3D simulation trajectory within $\Omega \in [0, 48]^3$ in $t \in [0, 750](s)$, using a 4th-order Runge–Kutta time integration method (Ren et al., 2022) under the periodic condition. Here, we define $D_u = 0.2, D_v = 0.1, \alpha = 0.025, \beta = 0.055, \Delta t = 0.25(s)$ and $\Delta x = 2$.

**Black Sea Dataset.** The BS dataset [1] provides measured data of daily mean sea surface temperature $T$ and water flow velocities $\mathbf{u}$ on the Black Sea over several years. The collection of these data was completed by Euro-Mediterranean Center on Climate Change (CMCC) in Italy starting from June 1, 1993, to June 30, 2021, with a horizontal resolution of $1/27° \times 1/36°$.

---

[1] `https://data.marine.copernicus.eu/product/BLKSEA_MULTIYEAR_PHY_007_004/description`

Table S3: Summary analysis of Baselines.

| Model | Grid Domain | Irregular Domain | Kernel Netwotk | Spectrum Netwotk | Graph Netwotk | Attention Netwotk | Multi Scale |
|---|---|---|---|---|---|---|---|
| FNO | ✓ | ✗ | ✗ | ✓ | ✗ | ✗ | ✗ |
| GAT | ✓ | ✓ | ✗ | ✗ | ✓ | ✓ | ✗ |
| GATv2 | ✓ | ✓ | ✗ | ✗ | ✓ | ✓ | ✗ |
| MGN | ✓ | ✓ | ✗ | ✗ | ✓ | ✗ | ✗ |
| MP-PDE | ✓ | ✓ | ✗ | ✗ | ✓ | ✗ | ✗ |
| FFNO | ✓ | ✗ | ✗ | ✓ | ✗ | ✗ | ✗ |
| Geo-FNO | ✓ | ✓ | ✗ | ✓ | ✗ | ✗ | ✗ |
| Transolver | ✓ | ✓ | ✗ | ✗ | ✗ | ✓ | ✗ |
| Ours | ✓ | ✓ | ✗ | ✗ | ✓ | ✗ | ✗ |

## D.2 SUPPLEMENTARY DETAILS OF BASELINE

**Fourier Neural Operator (FNO).** The most promising spectral approach, FNO (Li et al., 2021a) proposed a neural operator in the Fourier domain to model dynamics. In this work, it was implemented for 2D and 3D spatial grid domains. The hyperparameters are taken from (Li et al., 2021a).

**Graph Attention Network (GAT).** GAT (Velickovic et al., 2017) proposed a graph network with a masked self-attention mechanism, aiming to address the shortcomings of graph convolutions.

**Graph Attention Network Variant (GATv2).** GATv2 (Brody et al., 2022) proposed a dynamic graph attention variant to remove the limitation of static attention in complex controlled problems, which is strictly more expressive than GAT.

**MeshGraphNet (MGN).** MGN (Pfaff et al., 2021) provided a type of neural network architecture designed specifically for modeling physical systems that can be represented as meshes or graphs. Specifically, its " Encoder-Processor-Decoder" architecture in (Pfaff et al., 2021) has been widely adopted in many supervised learning tasks, such as fluid and solid mechanics constrained by PDEs. Relevant parameters are referenced from (Pfaff et al., 2021).

**MP-Neural-PDE Solver (MP-PDE).** MP-PDE (Brandstetter et al., 2022) proposed the temporal bundling and push-forward techniques to encourage zero-stability in training autoregressive models. Relevant parameters are referenced from (Brandstetter et al., 2022).

**Factorized Fourier Neural Operator (FFNO).**

Factorized Fourier Neural Operator (FFNO) (Tran et al., 2021) factorizes the representation into separable Fourier representation to reduce the spatial and temporal complexity, improving its scalability. Relevant parameters are referenced from (Tran et al., 2021).

**Geometry-informed FNO (Geo-FNO).** Geometry-informed FNO (Geo-FNO) (Li et al., 2023b; 2024a) maps the irregular domain into a uniform grid, preserving the computation efficiency and handling the arbitrary geometries. Relevant parameters are referenced from (Li et al., 2023b).

**Transolver.** Transolver (Wu et al., 2024) proposed a new Physics Attention to adaptively split the discretized domain into a series of learnable slices of flexible shapes, effectively capture intricate physical correlations under complex geometrics. Relevant parameters are referenced from (Wu et al., 2024).

## D.3 SUPPLEMENTARY DETAILS OF EXPERIMENT

### D.3.1 SUMMARY OF PARAMETERS OF ALL MODELS

In this part, we list the parameters of all models on various datasets in Tables S4 to S12, including the number of layers, the learning rate, the hidden dimension, etc.

Table S4: Range of training hyperparameters for FNO

| Dataset | No. of Layers | Std. of Noise | Learning rate | Hidden dimension | Batch size | No. of Parameters |
|---|---|---|---|---|---|---|
| 2D Burgers | 4 | $1\times10^{-4}$ | $1\times10^{-4}$ | 128 | 100 | 541,436 |
| 2D FN | 4 | $1\times10^{-4}$ | $1\times10^{-4}$ | 128 | 100 | 541,436 |
| 2D GS RD | 4 | $1\times10^{-4}$ | $1\times10^{-4}$ | 128 | 100 | 554,930 |
| 3D GS RD | 4 | $1\times10^{-4}$ | $1\times10^{-4}$ | 128 | 10 | 515,746 |

Table S5: Range of training hyperparameters for GAT

| Dataset | No. of Layers | Std. of Noise | Learning rate | Hidden dimension | Batch size | No. of Parameters |
|---|---|---|---|---|---|---|
| 2D Burgers | 4 | $1\times10^{-4}$ | $1\times10^{-4}$ | 128 | 100 | 779,014 |
| 2D FN | 4 | $1\times10^{-4}$ | $1\times10^{-4}$ | 128 | 100 | 779,014 |
| 2D GS RD | 4 | $1\times10^{-4}$ | $1\times10^{-4}$ | 128 | 100 | 779,014 |
| 3D GS RD | 4 | $1\times10^{-4}$ | $1\times10^{-4}$ | 128 | 5 | 779,014 |
| 2D BS | 4 | $1\times10^{-2}$ | $1\times10^{-4}$ | 128 | 20 | 805,350 |

Table S6: Range of training hyperparameters for GATv2

| Dataset | No. of Layers | Std. of Noise | Learning rate | Hidden dimension | Batch size | No. of Parameters |
|---|---|---|---|---|---|---|
| 2D Burgers | 4 | $1\times10^{-4}$ | $1\times10^{-4}$ | 128 | 100 | 778,630 |
| 2D FN | 4 | $1\times10^{-4}$ | $1\times10^{-4}$ | 128 | 100 | 778,630 |
| 2D GS RD | 4 | $1\times10^{-4}$ | $1\times10^{-4}$ | 128 | 100 | 778,630 |
| 3D GS RD | 4 | $1\times10^{-4}$ | $1\times10^{-4}$ | 128 | 5 | 778,630 |
| 2D BS | 4 | $1\times10^{-2}$ | $1\times10^{-4}$ | 128 | 20 | 804,966 |

Table S7: Range of training hyperparameters for MGN

| Dataset | No. of Layers | Std. of Noise | Learning rate | Hidden dimension | Batch size | No. of Parameters |
|---|---|---|---|---|---|---|
| 2D Burgers | 4 | $1\times10^{-4}$ | $1\times10^{-4}$ | 128 | 100 | 514,236 |
| 2D FN | 4 | $1\times10^{-4}$ | $1\times10^{-4}$ | 128 | 100 | 514,236 |
| 2D GS RD | 4 | $1\times10^{-4}$ | $1\times10^{-4}$ | 128 | 100 | 514,236 |
| 3D GS RD | 4 | $1\times10^{-4}$ | $1\times10^{-4}$ | 128 | 5 | 514,236 |
| 2D BS | 4 | $1\times10^{-2}$ | $1\times10^{-4}$ | 128 | 20 | 540,572 |

Table S8: Range of training hyperparameters for MP-PDE

| Dataset | No. of Layers | Std. of Noise | Learning rate | Hidden dimension | Batch size | No. of Parameters |
|---|---|---|---|---|---|---|
| 2D Burgers | 4 | $1\times10^{-4}$ | $1\times10^{-4}$ | 128 | 100 | 528,658 |
| 2D FN | 4 | $1\times10^{-4}$ | $1\times10^{-4}$ | 128 | 100 | 528,658 |
| 2D GS RD | 4 | $1\times10^{-4}$ | $1\times10^{-4}$ | 128 | 100 | 528,658 |
| 3D GS RD | 4 | $1\times10^{-4}$ | $1\times10^{-4}$ | 128 | 5 | 528,658 |
| 2D BS | 4 | $1\times10^{-2}$ | $1\times10^{-4}$ | 128 | 20 | 554,994 |

### D.3.2 DATA SCALING TEST

We also have performed a data scaling test and report the results (data size vs. prediction error) in the Table S13 below to support our claim of low data requirement.

Table S9: Range of training hyperparameters for CeGNN

| Dataset | No. of Layers | Std. of Noise | Learning rate | Hidden dimension | Batch size | No. of Parameters |
|---|---|---|---|---|---|---|
| 2D Burgers | 4 | $1\times10^{-4}$ | $1\times10^{-4}$ | 128 | 100 | 1,482,674 |
| 2D FN | 4 | $1\times10^{-4}$ | $1\times10^{-4}$ | 128 | 100 | 1,482,674 |
| 2D GS RD | 4 | $1\times10^{-4}$ | $1\times10^{-4}$ | 128 | 100 | 1,482,674 |
| 3D GS RD | 4 | $1\times10^{-4}$ | $1\times10^{-4}$ | 128 | 5 | 1,482,674 |
| 2D BS | 4 | $1\times10^{-2}$ | $1\times10^{-4}$ | 128 | 20 | 1,509,010 |

Table S10: Range of training hyperparameters for FFNO

| Dataset | No. of Layers | Std. of Noise | Learning rate | Hidden dimension | Batch size | No. of Parameters |
|---|---|---|---|---|---|---|
| 2D Burgers | 4 | $1\times10^{-4}$ | $1\times10^{-4}$ | 128 | 100 | 1,072,388 |
| 2D FN | 4 | $1\times10^{-4}$ | $1\times10^{-4}$ | 128 | 100 | 1,072,388 |
| 2D GS RD | 4 | $1\times10^{-4}$ | $1\times10^{-4}$ | 128 | 100 | 1,072,388 |
| 3D GS RD | 4 | $1\times10^{-4}$ | $1\times10^{-4}$ | 128 | 10 | 1,334,660 |

Table S11: Range of training hyperparameters for Geo-FNO

| Dataset | No. of Layers | Std. of Noise | Learning rate | Hidden dimension | Batch size | No. of Parameters |
|---|---|---|---|---|---|---|
| 2D Burgers | 4 | $1\times10^{-4}$ | $1\times10^{-4}$ | 128 | 100 | 727,396 |
| 2D FN | 4 | $1\times10^{-4}$ | $1\times10^{-4}$ | 128 | 100 | 727,396 |
| 2D GS RD | 4 | $1\times10^{-4}$ | $1\times10^{-4}$ | 128 | 100 | 727,396 |
| 3D GS RD | 4 | $1\times10^{-4}$ | $1\times10^{-4}$ | 128 | 10 | 2,705,569 |
| 2D BS | 4 | $1\times10^{-2}$ | $1\times10^{-4}$ | 128 | 20 | 727,557 |

Table S12: Range of training hyperparameters for Transolver

| Dataset | No. of Layers | Std. of Noise | Learning rate | Hidden dimension | Batch size | No. of Parameters |
|---|---|---|---|---|---|---|
| 2D Burgers | 4 | $1\times10^{-4}$ | $1\times10^{-4}$ | 128 | 100 | 1,516,354 |
| 2D FN | 4 | $1\times10^{-4}$ | $1\times10^{-4}$ | 128 | 100 | 1,516,354 |
| 2D GS RD | 4 | $1\times10^{-4}$ | $1\times10^{-4}$ | 128 | 100 | 1,516,354 |
| 3D GS RD | 4 | $1\times10^{-4}$ | $1\times10^{-4}$ | 128 | 10 | 1,631,042 |
| 2D BS | 4 | $1\times10^{-2}$ | $1\times10^{-4}$ | 128 | 20 | 1,516,739 |

Table S13: Data scaling results (prediction error, RMSE) of CeGNN, MGN and MP-PDE on 2D Burgers equation.

| Model | No.of layers | Dimension | RMSE ↓ | | |
|---|---|---|---|---|---|
| | | | Data size = 10 | Data size = 20 | Data size = 30 |
| CeGNN | 4 | 128 | **0.00664** | **0.00413** | **0.00226** |
| MGN | 4 | 128 | 0.01174 | 0.00926 | 0.00636 |
| MP-PDE | 4 | 128 | 0.01784 | 0.01442 | 0.00911 |

### D.3.3 THE EFFICACY OF CELL FEATURES

In this part, we also have test the efficacy of cell features on the performance of graph-based baselines across all benchmarks with various methods and report the results with RMSE metrics in the Table S14 below to support our claim of introducing cells.

Table S14: Efficacy of cell features on the performance of graph-based baselines across all benchmarks with various methods.

| Model | RMSE ↓ | | | | |
|-------|--------|--------|--------|--------|--------|
| | **2D Burgers** $(\Delta t = 0.001s)$ | **2D FN** $(\Delta t = 0.002s)$ | **2D GS RD** $(\Delta t = 0.25s)$ | **3D GS RD** $(\Delta t = 0.25s)$ | **2D BS** $(\Delta t = 1day)$ |
| MGN | 0.01174 | 0.02108 | 0.02917 | 0.01925 | 0.61475 |
| MGN + Cell | 0.00826 | 0.00791 | 0.00832 | 0.00694 | 0.58019 |
| Promotion (%) ↑ | 29.6 | 62.4 | 71.4 | 63.9 | 5.6 |
| MP-PDE | 0.01784 | 0.02848 | 0.03860 | 0.06528 | 0.60761 |
| MP-PDE + Cell | 0.00951 | 0.01193 | 0.00947 | 0.00992 | 0.59313 |
| Promotion (%) ↑ | 46.7 | 58.1 | 75.4 | 84.8 | 2.38 |
| Ours | 0.00664 | 0.00364 | 0.00248 | 0.00138 | 0.55599 |

Table S15: Quantitative ablation results about the cell position information for CeGNN and other two cases under RMSE metrics.

| Model | RMSE ↓ | | | | |
|-------|--------|--------|--------|--------|--------|
| | **2D Burgers** $(\Delta t = 0.001s)$ | **2D FN** $(\Delta t = 0.002s)$ | **2D GS RD** $(\Delta t = 0.25s)$ | **3D GS RD** $(\Delta t = 0.25s)$ | **2D BS** $(\Delta t = 1day)$ |
| CeGNN | 0.00664 | 0.00364 | 0.00248 | 0.00138 | 0.55599 |
| w/o Cell Pos. | 0.00721 | 0.00477 | 0.00439 | 0.00274 | 0.56098 |
| w Cell (Pos. to B) | 0.00720 | 0.00490 | 0.00445 | 0.00276 | 0.56040 |
| MGN | 0.01174 | 0.02108 | 0.02917 | 0.01925 | 0.61475 |

Table S16: Summary of CeGNN, MGN, CCNNs, and MPSNs

| Model | Message Level | Special Complex Predefinition | Simplex, Complex Type | Application |
|-------|---------------|-------------------------------|----------------------|-------------|
| CeGNN | 2 | No | 3 | Dynamics |
| MGN | 2 | No | 2 | Dynamics |
| CCNNs | 3 | No | 3 | Classification |
| MPSNs | 2 | Yes | 5 | Classification |

#### D.3.4    DISCUSSION OF THE POSITION INFORMATION IN CELLS

In this part, we provide other two cases "CeGNN w/o Cell Pos." and " w Cell (Pos. to B)" in the following Table S15, where " w/o Cell Pos." represents the cell initial feature without the position awareness, and " w Cell (Pos. to B)" is replacing the distance to the cell center with the distance to the nearest PDE boundary. The results in Table S15 show that the performance improvement of CeGNN is not only due to add position awareness into the cell initial feature, but also the second-order refinement of the discrete space.

#### D.3.5    DISCUSSION WITH OTHER HIGHER-ORDER GRAPHS

In this part, we mainly focus on discussing the excellent works CCNNs in (Hajij et al., 2022) and MPSNs in (Bodnar et al., 2021). A summary of these models is shown in the following Table S16.

Firstly, MGN achieves message passing through a two-level structure ($edge \rightarrow node$). On this basis, CCNNs in (Hajij et al., 2022) leverages combinatorial complexes to achieve message passing through a three-level structure ($cell \rightarrow edge \rightarrow node$). Meanwhile, MPSNs in (Bodnar et al., 2021) performs message passing through a two-level structure ($complexes \rightarrow simplex$) on various simplicial complexes (SCs), which primarily include one simplex (node) and four types of complexes with varying adjacent simplices (e.g., boundary adjacencies, co-boundary adjacencies, lower-adjacencies, and upper-adjacencies), to enhance feature distinguishability and thus improve classification performance.

Table S17: Comparison result between two-level and three-level mechanisms

| Model | RMSE ↓ | | | | |
|---|---|---|---|---|---|
| | 2D Burgers ($\Delta t = 0.001s$) | 2D FN ($\Delta t = 0.002s$) | 2D GS RD ($\Delta t = 0.25s$) | 3D GS RD ($\Delta t = 0.25s$) | 2D BS ($\Delta t = 1day$) |
| CeGNN | 0.00664 | 0.00364 | 0.00248 | 0.00138 | 0.55599 |
| MGN (2-level) | 0.01174 | 0.02108 | 0.02917 | 0.01925 | 0.61475 |
| MGN (3-level) | 0.03220 | 0.04196 | 0.05884 | 0.08218 | 0.61524 |

In contrast, CeGNN sets apart from these works and adopts a novel two-level structure ($[cell, edge] \rightarrow node$) from the spatial perspective, which is more suitable to learn implicit dynamic mechanism.

Since there are only node labels for the supervised learning, the three-level message passing mechanism in (Hajij et al., 2022) poses significant training challenges. However, our two-level message passing sequence reduces the high coupling degree in (Hajij et al., 2022) and avoids the limitation for additional predefined special complexes (e.g., some simplicial complexes in (Bodnar et al., 2021) ). A comparison result between two-level and three-level mechanisms is shown in the following Table S17. The results in Table S17 demonstrate that three-level mechanism underperforms than two-level mechanism in our all cases.

