# OpenReview forum: "Spatiotemporal Learning on Cell-embedded Graphs"
_ICLR.cc/2025/Conference — Submitted to ICLR 2025_

### Official Review · Reviewer_KT6u · 2024-10-28

**Soundness:** 1
**Presentation:** 3
**Contribution:** 3
**Rating:** 5
**Confidence:** 4

**Summary:**

This paper proposes an end-to-end graph-based framework called CeGNN to address limitations of existing Graph Neural Networks in learning complex spatiotemporal dynamics, particularly the over-smoothing issue, and aims to enhance prediction accuracy and generalization ability. The authors introduce two key components: Cell-embedded MPNN block and Feature-Enhanced (FE) block. Through experiments on several PDE systems, the paper demonstrates that CeGNN outperforms other baseline models.

**Strengths:**

- The paper is well-organized and easy to follow.

- The authors present abundant experimental results and visualizations to validate their ideas.

- CeGNN achieves superior performance compared to the compared methods.

**Weaknesses:**

- The baseline methods are relatively weak. The authors did not include recent advancements in the field from 2023-2024, raising concerns about the effectiveness of the proposed method.

- Modeling with higher-order graphs is a widely studied topic. Can the authors more explicitly summarize the contributions of CellMPNN compared to existing approaches?

- The paper lacks a theoretical discussion on the effectiveness of CeGNN. Can the authors discuss the source of CeGNN's effectiveness from a theoretical perspective?

- FE modules are not clearly defined.

**Questions:**

Please address the weaknesses mentioned above.

---

> ### Author Response · Authors · 2024-11-21
> **Reply to Reviewer KT6u (Part 1)**
>
> Thanks for your constructive comments!
>
> >**Weakness 1:** Add recent works as new baselines.
>
> **Reply:** Great suggestion! We have added results of comparison with several other baseline models in the revised paper (see Subsection 4.3 on Page 7, Table 1 on Page 9, and Appendix Tables S3, S10-S12 on Pages 19-21), namely,
> - The additional experimental results are added in **Table 1**.
> - A discussion about the baseline comparision results is added in **Subsection 4.3**.
> - Details of the new baselines, e.g., the hyperparameters, have been added in **Appendix Subsection D.2, Appendix Tables S3, S10-S12**.
>
> Specifically, we have considered three additional baselines (aka, FFNO, Geo-FNO, Transolver) to compare with the performance of CeGNN on our all benchmarks. A summary of these models is shown in the following **Table F**. The results show that Geo-FNO [1] and Transolver [3] perform poorly on the limited training dataset, which is significantly smaller, by at least an order of magnitude less, than the hundreds or thousands of datasets typically used for such models. We have added the new results to our revised paper, referring to **Table 1** (Page 9).
>
> **Table F: RMSE metrics for CeGNN and other models.**
> | Model         | 2D Burgers | 2D FN   | 2D GS   | 3D GS            | 2D BS   |
> | ------------- | ---------- | ------- | ------- | ---------------- | ------- |
> | CeGNN         | **0.0066**    | **0.0036** | **0.0024** | **0.0013**          | **0.5560** |
> | Geo-FNO [1]    | 0.5936    | 20.514  | 0.1867 | NaN              | 1.2893  |
> | FFNO [2]       | 0.0334    | 0.1192 | 0.0362 | 0.0359 |  -       |
> | Transolver [3] | 0.1742    | 0.1372 | 0.1859 | 0.1520          | 0.8199 |
>
> **References:**
>
> [1] Li, et al. A. Fourier neural operator with learned deformations for pdes on general geometries. arXiv:2207.05209, 2022.
>
> [2] Tran, et al. Factorized fourier neural operators. In ICLR, 2023.
>
> [3] Wu, et al. Transolver: A fast transformer solver for pdes on general geometries. In ICML, 2024.
>
>
> >**Weakness 2a:** Summarize the contributions of CellMPNN block from a theoretical perspective.
>
> **Reply:** Great comment! Generally, the traditional message passing (MP) mechanism can be regarded as a refinement on a discrete space, analogous to an interpolation operation, which implies that edges are essentially interpolated from nodes. A MP mechanism introducing the cell further enhances the refinement of the discrete space (secondary refinement), thereby reducing the magnitude of discretization errors spatially, paving the way for its application in complex graph structures. Please see below **Definition 1** on cell in graph and **Corollary 1** on expressive power of cell-embedded MP. Therefore, we proposed a new two-level cell-embedded mechanism to process the message on graphs, which forms the key novelty of the proposed model.
>
> ***Definition 1 (Cell in Graph):*** Let $G = (V, E)$ be a graph, where $V$ is the set of nodes $\mathbf{v}$ and $E \subseteq V \times V$ is the set of edges. A cell in $G$ is a subset of nodes $C \subseteq V$, such that the nodes in $C$ form a complete subgraph (clique) or satisfy predefined structural relationships. In particular, a $k$-cell $C_k$ in a graph $G$ contains $k+1$ nodes, where $\forall i, j \in C_k$, $(\mathbf{v}_i, \mathbf{v}_j) \in E$, representing various structures, such as node ($k=0$), edge ($k=1$), triangle ($k=2$), tetrahedron ($k=3$), and so on.
>
> ***Corollary 1 (Expressive Power):*** Given a graph $G$ including many $k$-cell ($k=0,1,2,\dots$), there exists a cell-based scheme that is more expressive than Weisfeiler-Lehman (WL) tests in distinguishing non-isomorphic graphs (see the proof in **Appendix Subsection C.2** of the revised paper (Pages 16-17)).

---

> ### Author Response · Authors · 2024-11-21
> **Reply to Reviewer KT6u (Part 2)**
>
> >**Weakness 2b:** Summarize the contributions of CellMPNN block from an experimental verification perspective.
>
> **Reply:** In particular, a summary of CellMPNN block on the performance of some graph-based baselines across all benchmarks is shown in the following **Table H**. The results in **Table H** show  the positive effect of CellMPNN block. We have added the new results to our revised paper, referring to **Appendix Table S14** in the revised paper (Page 22).
>
> **Table H: Summary of CellMPNN block on the performance of graph-based baselines across all benchmarks**
> | Model         | 2D Burgers | 2D FN   | 2D GS   | 3D GS   | 2D BS   |
> | ------------- | ---------- | ------- | ------- | ------- | ------- |
> | MGN           | 0.0117    | 0.0210 | 0.0291 | 0.0192 | 0.6147 |
> | MGN + Cell    | 0.0082    | 0.0079        | 0.0083        |   0.0069 | 0.5801 |
> | Promotion (%) | 29.6      |   62.4      |  71.4       |        63.9 |  5.6       |
> | MP-PDE        | 0.0178    | 0.0284 | 0.0386 | 0.0652 | 0.6076 |
> | MP-PDE + Cell | 0.0095   |  0.0119       | 0.0094        |  0.0099       | 0.5931        |
> | Promotion (%) | 46.7          |  58.1       | 75.4        |      84.8   |   2.38      |
> | CeGNN         | **0.0066**    | **0.0036** | **0.0024** | **0.0013** | **0.5560** |
>
> In our study, we also attempted to enhance feature distinguishability through a **three-level** update structure ($cell \rightarrow edge \rightarrow node$). Since there are only node labels for the supervised learning, this three-level message passing mechanism failed to yield performance lift which might be due to the introduction of redundant path of information passing. Therefore, we adapt two-level message passing sequence to reduce the high coupling degree and avoid the limitation for additional predefined special complexes. A comparison result between two-level and three-level mechanisms is shown in the following **Table G**. The results in **Table G** demonstrate that the three-level mechanism underperforms compared with the proposed two-level mechanism in all cases. We have added the new results to our revised paper. Please refer to **Appendix Table S17** in the revised paper (Page 23).
>
> **Table G: Comparison result between two-level and three-level mechanisms**
> | Model         | 2D Burgers | 2D FN   | 2D GS   | 3D GS   | 2D BS   |
> | ------------- | ---------- | ------- | ------- | ------- | ------- |
> | CeGNN         | **0.0066**    | **0.0036** | **0.0024** | **0.0013** | **0.5559** |
> | MGN (2-level) | 0.0117    | 0.0210 | 0.0291 | 0.0192 | 0.6148 |
> | MGN (3-level) | 0.0322    | 0.0419 | 0.0588 | 0.0821 | 0.6152 |

---

> ### Author Response · Authors · 2024-11-21
> **Reply to Reviewer KT6u (Part 3)**
>
> >**Weakness 3:** Unclearly defined FE modules.
>
> **Reply:** Great comment! Following your suggestion, we have added detailed explanation in the revised paper (see Section 3.1.1 on Pages 4-5 and Appendix Section C.3 on Page 17).
>
> - We update the **FE block in Figure 2** to further explain its process and redefine **Algorithm 1** (placed in Appendix).
> - A primary theoretical deduction is updated in **Subsubsection 3.1.1**.
> - More theoretical knowledge have also been added in **Appendix Section C.3**.
>
> Concretely, the FE block is inspired by interaction models in physics and mathematically hypothesized to capture nonlinear dependencies, enhancing the model's representation power.
>
> ***Hypothesis.*** The FE block's hypothesis could be framed as: *By capturing second-order interactions between latent features and applying selective filtering, the model can better represent complex structures or relationships in the data*.
>
> ***Physical Analogy.*** In physics, the outer product and second-order terms are often used to model interactions, such as stress tensors in mechanics or pairwise correlations in quantum mechanics. Here, the module could draw an analogy to systems where interactions between individual components (features) are crucial to the overall behavior.
>
> ***Process Overview.*** In detail, it regards the node latent feature $\overline{\mathbf{h}}_ i \in\mathbb{R}^{D\times 1}$ as basis and builds a higher-order tensor feature $\mathbf{H}_ {i} \in\mathbb{R}^{D\times D}$ via an outer-product operation, e.g., $\overline{\mathbf{h}}_ {i}\otimes\overline{\mathbf{h}}_ {i}$. This process creates abundant second-order nonlinear terms to enrich the feature map. We then use a mask operation with $\mathbf{M}\in\mathbb{R}^{D\times D}$ to randomly sample these terms, filtering the appropriate information by a learnable weight tensor $\mathbf{W}\in\mathbb{R}^{D\times D\times D}$ to enhance the model's representation capacity.
>
> ***Outer Product as Basis Expansion.*** The outer product operation $\otimes$ on the reshaped feature map $\overline{\mathbf{h}}_ {i}\in\mathbb{R}^{D\times 1}$ expands the original latent feature space into a higher-order tensor space. This expansion introduces second-order terms (e.g., $\alpha\beta$ for $\alpha, \beta \in \overline{\mathbf{h}}_{i}$), which can capture interactions between individual components of the original feature $\mathbf{h} _{i} \in\mathbb{R}^{D}$. Mathematically, the second-order tensor reads $\overline{\mathbf{h}} _{i} \otimes\overline{\mathbf{h}} _{i}$.This operation creates a richer feature map with cross-term interactions that may not be explicitly encoded in the original latent space.
>
> ***Lemma 1 (Nonlinear Representation):*** The second-order terms $\alpha\beta$ can model nonlinear dependencies between features. This is particularly useful for capturing complex interactions that linear transformations (e.g., simple dot products) might overlook.
>
> ***Definition 2:*** The FE block expands the latent feature $\overline{\mathbf{h}}_ i \in\mathbb{R}^{D\times 1}$ of node $i$ into a higher-order tensor space using an outer product: $\mathbf{H}_ {i}=\overline{\mathbf{h}}_ {i} \otimes\overline{\mathbf{h}}_ {i}$, where $\mathbf{H}_ {i}\in\mathbb{R}^{D\times D}$ is a higher-order feature map.
>
> ***Regularization via Masking.*** Masking introduces sparsity in $\mathbf{H}_ {i}$, reducing overfitting. If $\mathbf{M}_ {jk}$ is selected, $\mathbf{M}_ {jk} =1$. Otherwise, $\mathbf{M}_ {jk} =0$. Here $j$ and $k$ index the $M \in\mathbb{R}^{D\times D}$ components.
>
> Theoretically, masking operation serves two purposes: (1) Reducing computational complexity by randomly sampling terms from the higher-order feature space. (2) Regularizing the model by introducing sparsity, which can prevent overfitting in high-dimensional spaces.
>
> ***Learnable Filtering.*** The learnable weight tensor $\mathbf{W}\in\mathbb{R}^{D\times D\times D}$ acts as a filter, selecting and emphasizing the most informative terms.
>
> ***Definition 3:*** A mask operation $\mathbf{M} \in \mathbb{R}^{D\times D}$ is applied to randomly sample elements in $\mathbf{H}_ {i}$, and the resulting masked tensor is processed using a learnable weight tensor $\mathbf{W} \in \mathbb{R}^{D\times D\times D}$ as follows: $\tilde{\mathbf{h}}_ i$ = ($\mathbf{M} \odot\mathbf{H}_ {i}): \mathbf{W}$, where $\odot$ represents element-wise multiplication and $:$ denotes double contraction of tensors. The resulting feature $\tilde{\mathbf{h}}_ {i} \in\mathbb{R}^{1\times D}$ enriches the representation of $\mathbf{h}_ {i} \in \mathbb{R}^{D}$.
>
> ***Corollary 2 (Representation Power):*** The full feature map $\mathbf{H}_i$ contains $D^2$ terms for a $D$-dimensional input feature vector $\mathbf{h}_i$. After masking, the effective representation space reduces by the sparsity of $\mathbf{M}$. The learnable filter $\mathbf{W}$ further narrows this down to the most critical terms.

---

> ### Author Response · Authors · 2024-11-21
> **Reply to Reviewer KT6u (Part 4)**
>
> >**Weakness 4:** Discuss the CeGNN's effectiveness from a theoretical perspective.
>
> **Reply:** Great suggestion! In fact, the novelty of CeGNN includes the two-level cell-embedded message passing mechanism and the unique feature-enhanced (FE) block. Given the cell-embedded mechanism that better captures the spatial dependency and the FE block that further learns the distinguishable features, CeGNN achieves superior performance compared with other baseline models across all benchmark datasets, as shown in **Table 1** in the revised paper (Page 9). Moreover, as shown in **Table 3** in the revised paper (Page 9), we have provided an ablation study on all benchmarks to assess the contributions of the FE and CellMPNN blocks in CeGNN.
>
> Following your suggestion, we have added additional discussions in the revised paper (see Subsection 3.1 on Pages 4-6, Appendix Section C on Pages 16-17), namely,
>
> - A primary theoretical deduction of FE block was added in **Subsubsection 3.1.1**.
> - A primary theoretical deduction of CellMPNN block was added in **Subsubsection 3.1.2**.
> - Theoretical preliminaries have also been added in **Appendix Section C**.
>
> In summary, we proposed a new two-level cell-embedded mechanism to process the message on graphs. Please also see our reply to **Weakness 2a and 2b** about the theoretical and experimental verification perspective of the effectiveness. Moreover, the FE block is to capture nonlinear dependencies, enhancing the model's representation power. More detailed theoretical discussion about FE block can be found in our reply to **Weakness 3**.
>
> Overall, we appreciate your constructive comments and suggestions. Please let us know if you have any other questions. We look forward to your feedback!

---

> ### Author Response · Authors · 2024-11-23
> **Looking forward to your feedback**
>
> Dear Reviewer KT6u,
>
> Again, thanks for your constructive comments. We would like to follow up on our rebuttal to ensure that all concerns have been adequately addressed. If there are any further questions or points that need discussion, we will be happy to address them. Your feedback is invaluable in helping us improve our work, and we eagerly await your response.
>
> Thank you very much for your time and consideration.
>
> Best regards,
>
> The Authors

---

> ### Author Response · Authors · 2024-11-25
> **Request your feedback before the end of the discussion period**
>
> Dear Reviewer KT6u:
>
> As the author-reviewer discussion period will end soon, we would appreciate it if you could review our responses at your earliest convenience. If there are any further questions or comments, we will do our best to address them before the discussion period ends.
>
> Thank you very much for your time and efforts. Looking forward to your response!
>
> Sincerely,
>
> The Authors

---

> ### Author Response · Authors · 2024-11-26
> **Sincerely looking forward to your feedback**
>
> Dear Reviewer KT6u,
>
> Again, thanks for your constructive comments, which are very much helpful for improving our paper. If there are any further questions or points that need discussion, we will be happy to address them. Your feedback is invaluable in helping us improve our work, and we eagerly await your response.
>
> Moreover, we have thoroughly proofread our paper, corrected typos and grammar mistakes, and re-organized the contents to improve the clarity of the paper. We believe the presentation has been **substantially improved** (see revisions marked in red color). Please refer to the **updated .pdf file**.
>
> Thank you very much for your time and consideration.
>
> Best regards,
>
> The Authors

---

> ### Comment · Reviewer_KT6u · 2024-11-27
>
> I appreciate the authors' response. However, I still lean to rejecting this paper for the following reasons:
>
> - As Reviewer rgEv pointed out, the comparison baselines are still outdated. Most of the methods being compared are from 2021-2023, which raises doubts about whether this truly demonstrates the superiority of the proposed method.
>
> - The credibility of the comparisons is questionable. The newly added GeoFNO (2023) crashes in most experiments, while Transolver (2024) performs significantly worse than MGN. I'm uncertain about the value of these comparisons. It either suggests that these baseline methods are not meaningful for comparison, or the authors may have implemented the baselines incorrectly.
>
> - While we should be cautious in our assessment, I still find that this paper lacks novelty in its contribution.

---

> ### Author Response · Authors · 2024-11-27
> **Reply to the additional comments from Reviewer KT6u**
>
> Thank you for your feedback. Your time and effort placed on reviewing our paper are greatly appreciated!
>
> > **Comment: Regarding the alleged "outdated" baselines (2021-2023).**
>
> **Reply:** We appreciate but respectfully *disagree* with the comment that the baselines used in our study are "outdated". In fact, we carefully selected a range of baselines (**eight in total**) that are widely recognized in the field (e.g., MGN [1] (ICLR, 2021), GAT [2] (arXiv, 2017), GATv2 [3] (arXiv, 2022), FNO [4] (ICLR, 2021)) and some latest representative methods (e.g., MP-PDE [5] (ICLR, 2022), FFNO [6] (ICLR, 2023), Geo-FNO [7] (NIPS, 2024), Transolver [8] (ICML, 2024)) for comparison. Many of these have been recently published in prestigious venues and actively used as representative baseline models in recent research. Hence, we believe that the baseline comparisons are up to date in the field of spatiotemporal dynamics prediction.
>
> Meanwhile, we have provided **extensive** experiments (see **Tables 1-5, Appendix Tables S13-S17** in the revised paper) comparing our method with all baselines to demonstrate our model's effectiveness. If the reviewer believes that there are more recent or more appropriate baselines we should include, we sincerely welcome your suggestions and would be more than happy to include them in final version of the paper. Thanks!
>
>
> > **Comment: Regarding the credibility of the comparisons and implementation concerns.**
>
> **Reply:** In fact, the underperformance of GeoFNO [7] (NIPS, 2024) and Transolver [8] (ICML, 2024) on limited training datasets likely stem from their *mapping techniques between the irregular mesh and the regular grid*, which rely on diverse training data. This is a *quite common* issue when these types of models are trained in small data regimes like the case we are specifically considering in this paper.
>
> Moreover, we would like to draw your attention that all the baselines were implemented with the utmost care, following publicly available codebases and the instructions provided in the corresponding papers. We ensure you that these implementation are correct, where the baseline comparison codes will be released along with our codes after the peer review stage.
>
> Our experiments (see **Table 1** in the revised paper) aim to evaluate all the methods across diverse settings. The results highlight the robustness of our model in scenarios where other baselines may struggle, particularly when the training datasets are limited. More detailed information of these models have been provided in **Appendix Subsection D.2, Appendix Tables S4-S12** in the revised paper.
>
> Hope this clarifies your concern.
>
>
> > **Comment: Regarding the novelty of our contribution.**
>
> **Reply:** In fact, the novelty of our proposed CeGNN model includes the introduction of two key modules in the network architecture, namely,
>
> - (1) ***the two-level cell-embedded message passing mechanism***, which better captures the spatial dependency;
> - (2) ***the unique feature-enhanced (FE) block***, which further learns the distinguishable features.
>
> With extensive experiments on various spatiotemporal systems, CeGNN achieves superior performance compared with other baseline models across all benchmark datasets, as shown in **Table 1** in the revised paper (Page 9). This has been thoroughly discussed and clearly demonstrated from the theoretical perspective (see **Subsubsection 3.1.2, Appendix Subsection C.1, C.2, D.3** in the revised paper) and evidence of the empirical results (see **Tables 1, 3 and 4, Appendix Tables S13-S17** in the revised paper). Hope this clarifies your concern.
>
> We also recognize that *novelty can sometimes be nuanced*, sometimes depending on reader's personal judgement. However, our proposed network architeture, consisting of the cell-embedded message passing mechanism and the FE block, is unique and new, with superior performance clearly contributing to its novelty.
>
> **Concluding Remark:** We appreciate the reviewer’s additional comments and critical evaluation. We sincerely hope to have your re-evaluation of our paper in light of our clarifications and contributions. Your possible consideration of updating the score is highly appreciated!
>
> We look forward to your feedback!
>
> ***References:***
>
> [1] Pfaff et al. Learning mesh-based simulation with graph networks. In ICLR, 2021.
>
> [2] Veličković et al. Graph attention networks. arXiv 2017.
>
> [3] Brody S, Alon U, Yahav E. How attentive are graph attention networks? arXiv 2022.
>
> [4] Li et al. Fourier neural operator for parametric partial differential equations. In ICLR, 2021.
>
> [5] Brandstetter et al. Message passing neural PDE solvers. In ICLR, 2022.
>
> [6] Tran et al. Factorized fourier neural operators. In ICLR, 2023.
>
> [7] Li et al. Geometry-informed neural operator for large-scale 3d pdes. In NIPS, 2024.
>
> [8] Wu et al. Transolver: A fast transformer solver for pdes on general geometries. In ICML, 2024.

---

### Official Review · Reviewer_1r7X · 2024-10-28

**Soundness:** 3
**Presentation:** 3
**Contribution:** 2
**Rating:** 5
**Confidence:** 3

**Summary:**

The paper presents a new model, the Cell-Embedded Graph Neural Network (CeGNN), for simulating spatiotemporal dynamics across different physical domains. CeGNN introduces learnable cell attributions to the traditional node-edge message-passing process, upgrading it to a higher-order scheme that captures volumetric information and improves spatial dependency learning. Additionally, the Feature-Enhanced (FE) block enriches feature representations, tackling the over-smoothness issue common in Graph Neural Networks (GNNs). Extensive experiments demonstrate that CeGNN achieves superior performance and generalization in predicting physical dynamics, particularly for Partial Differential Equations (PDEs) and real-world datasets.

**Strengths:**

Good empirical results.

**Weaknesses:**

The main weakness is lack of novelty. The main idea of this paper is the proposal of cell-attribution. In the field of topological learning, there have been several prior works proposing the idea of higher-order message passing, cell / simplicial complex neural networks. Please check the following literature:

(1) Topological Deep Learning: Going Beyond Graph Data (this is a great survey of Topological Deep Learning)
https://arxiv.org/abs/2206.00606

(2) Cell Complex Neural Networks
https://arxiv.org/abs/2010.00743

(2) Weisfeiler and Lehman Go Topological: Message Passing Simplicial Networks (for simplical complexes)
https://arxiv.org/abs/2103.03212

However, application of these topological methods in the domain of learning physical systems is new.

**Questions:**

Please address the weakness in novelty that I have raised!

---

> ### Author Response · Authors · 2024-11-21
> **Reply to Reviewer 1r7X (Part 1)**
>
> Thanks for your constructive comments!
>
> >**Weakness 1a:** The concern on novelty of cell-attribution.
>
> **Reply:** Thanks for your comment. In fact, the novelty of our proposed CeGNN model includes the introduction of two key modules in the network architecture, namely, (1) the two-level cell-embedded message passing mechanism and (2) the unique feature-enhanced (FE) block. Given the cell-embedded mechanism that better captures the spatial dependency and the FE block that further learns the distinguishable features, CeGNN achieves superior performance compared with other baseline models across all benchmark datasets, as shown in **Table 1** in the revised paper (Page 9).
>
> However, following your suggestion, we have added many contents in the revised paper (see Subsection 3.1 on Pages 4-6 and Appendix Tables S16, S17 on Page 23).
> - A primary theoretical deduction is added in **Subsection 3.1**.
> - Two experimental results are added in **Appendix Tables S16 and S17**.
> - More discussion have also been added in **Appendix Subsubsection D.3.5**.
>
> Generally, the traditional message passing (MP) mechanism can be regarded as a refinement on a discrete space, analogous to an interpolation operation, which implies that edges are essentially interpolated from nodes. A MP mechanism introducing the cell further enhances the refinement of the discrete space (e.g., secondary refinement), thereby reducing the magnitude of discretization errors spatially, paving the way for its application in complex graph structures. Please see below **Definition 1** on cell in graph and **Corollary 1** on expressive power of cell-embedded MP. Therefore, we proposed a new two-level cell-embedded mechanism to process the message on graphs, which forms the key novelty of the proposed model.
>
> ***Definition 1: (Cell in Graph)*** Let $G = (V, E)$ be a graph, where $V$ is the set of nodes $\mathbf{v}$ and $E \subseteq V \times V$ is the set of edges. A cell in $G$ is a subset of nodes $C \subseteq V$, such that the nodes in $C$ form a complete subgraph (clique) or satisfy predefined structural relationships. In particular, a $k$-cell $C_k$ in a graph $G$ contains $k+1$ nodes, where $\forall i, j \in C_k,$, $(\mathbf{v}_i, \mathbf{v}_j) \in E$, representing various structures, such as node ($k=0$), edge ($k=1$), triangle ($k=2$), tetrahedron ($k=3$), and so on.
>
> ***Corollary 1: (Expressive Power)*** Given a graph $G$ including many $k$-cell ($k=0,1,2,\dots$), there exists a cell-based scheme that is more expressive than Weisfeiler-Lehman (WL) tests in distinguishing non-isomorphic graphs (see the proof in **Appendix Subsection C.2** of the revised paper (Pages 16-17)).

---

> ### Author Response · Authors · 2024-11-21
> **Reply to Reviewer 1r7X (Part 2)**
>
> >**Weakness 1b:** Differences compared with exisiting topological graph learning methods.
>
> **Reply:** Based on the references mentioned by you, we focus on discussing CCNNs [1] and MPSNs [3], since the work in [2] shares similarity with CCNNs [1]. A summary of these models is shown in the following **Table D**. We have added this new table to our revised paper, referring to **Appendix Table S16** in the revised paper (Page 22).
>
> **Table D: Summary of CeGNN, MGN, and the works in [1] and [3].**
> | Model| Message Level | Special Complex Predefinition |Simplex and Complex Type | Application    |
> | - | - | - | - | - |
> | CeGNN    | 2 | No | 3 | Dynamics |
> | MGN      | 2 | No | 2 | Dynamics |
> | CCNNs[1] | 3 | No | 3 | Classification |
> | MPSNs[3] | 2 | Yes | 5 | Classification |
>
> Firstly, MGN achieves message passing through a **two-level** structure ($edge \rightarrow node$). On this basis, CCNNs in [1] leverages combinatorial complexes to achieve message passing through a **three-level** structure ($cell \rightarrow edge \rightarrow node$). Meanwhile, MPSNs in [3] performs message passing through a **two-level** structure ($complexes \rightarrow simplex$) on various simplicial complexes (SCs), which primarily include **one simplex** (node) and **four types of complexes** with varying adjacent simplices to enhance feature distinguishability and thus improve classification performance.
>
> In contrast, our CeGNN model sets apart from these works and adopts a **novel two-level** structure ($[cell, edge] \rightarrow node$) from the spatial perspective, which is more suitable to learn the underlying spatiotemporal dynamics. Since there are only node labels for the supervised learning, the three-level message passing mechanism in [1] failed to yield performance lift which might be due to the introduction of redundant path of information passing. However, our two-level message passing sequence reduces the high coupling degree in [1] and avoids the limitation for additional predefined special complexes (e.g., some simplicial complexes) in [3]. A comparison result between two-level and three-level mechanisms is shown in the following **Table E**. The results in Table E demonstrate that the three-level mechanism underperforms compared with our proposed two-level mechanism in all cases. We have added the new results to our revised paper, referring to **Appendix Table S17** in the revised paper (Page 23). Moreover, as shown in **Table 3** in the revised paper (Page 9), we have provided an ablation study on all benchmarks to assess the contributions of the FE and CellMPNN blocks in CeGNN.
>
> **Table E: Comparison result between two-level and three-level mechanisms**
> | Model | 2D Burgers | 2D FN | 2D GS | 3D GS | 2D BS|
> | - | - | - | - | - | -|
> | CeGNN | **0.0066** | **0.0036** | **0.0024** | **0.0013** | **0.5559** |
> | MGN (2-level) | 0.0117 | 0.0210 | 0.0291 | 0.0192 | 0.6147 |
> | MGN (3-level) | 0.0322 | 0.0419 | 0.0588 |0.0821 |  0.6152       |
>
> **References:**
>
> [1] Hajij, et al. Topological deep learning: Going beyond graph data. arXiv:2206.00606, 2022.
>
> [2] Hajij, et al. Cell complex neural networks. arXiv:2010.00743, 2020.
>
> [3] Bodnar, et al. Weisfeiler and lehman go topological: Message passing simplicial networks, In ICML, 2021.
>
> We appreciate your constructive comments and suggestions. Please let us know if you have any other questions. We look forward to your feedback!

---

> ### Author Response · Authors · 2024-11-23
> **Looking forward to your feedback**
>
> Dear Reviewer 1r7X,
>
> Again, thanks for your constructive comments. We would like to follow up on our rebuttal to ensure that all concerns have been adequately addressed. If there are any further questions or points that need discussion, we will be happy to address them. Your feedback is invaluable in helping us improve our work, and we eagerly await your response.
>
> Thank you very much for your time and consideration.
>
> Best regards,
>
> The Authors

---

> > ### Comment · Reviewer_1r7X · 2024-11-27
> > **Keep my score**
> >
> > Dear Authors,
> >
> > Thank you for submitting the rebuttal!
> >
> > I did not raise concern about experiments, but on the theoretical front. I am still not convinced that your work is theoretically different from other topological models such as models learning on simplicial complexes or cell complexes. Simple propagation (between cell - edge - node) as in your work is indeed just a special case of the general topological model. I will keep my score. Please check the work of TopoX: https://github.com/pyt-team/TopoModelX. I believe your model can be implemented conveniently given this framework.
> >
> > Best,
> > Reviewer

---

> > > ### Author Response · Authors · 2024-11-27
> > > **Reply to additional comment from Reviewer 1r7X**
> > >
> > > Thank you for your additional feedback. Your time and effort placed on reviewing our paper are greatly appreciated!
> > >
> > > **Comment:** Regarding the concern of theoretical contribution.
> > >
> > > **Reply:** We appreciate your concern regarding the theoretical novelty of our approach. In fact, the novelty of our proposed CeGNN model includes the introduction of two key modules in the network architecture, namely,
> > >
> > > - (1) ***the two-level cell-embedded message passing mechanism***, which better captures the spatial dependency;
> > > - (2) ***the unique feature-enhanced (FE) block***, which further learns the distinguishable features.
> > >
> > > With extensive experiments on various spatiotemporal systems, CeGNN achieves superior performance compared with other baseline models across all benchmark datasets, as shown in **Table 1** in the revised paper (Page 9).
> > >
> > > For your comment, we respectfully believe that it overlooks a key distinction in our approach, specifically **the integration of the FE block**, which introduces additional complexity and novelty to our model. The efficacy of FE blcok has been thoroughly discussed from the theoretical perspective (see **Subsubsection 3.1.1, Appendix Subsection C.3** in the revised paper) and clearly demonstrated by evidence of the empirical results (see **Tables 2-3** in the revised paper). Specially, experimental results in **Table A** show the efficacy of FE block, referring to the **Table 2** in the revised paper.
> > >
> > > **Table A: Efficacy of feature-enhanced (FE) block.**
> > > | Model         | 2D Burgers | 2D FN   | 2D GS   | 3D GS   | 2D BS   |
> > > | ------------- | ---------- | ------- | ------- | ------- | ------- |
> > > | MGN           | 0.0117     | 0.0210  | 0.0291  | 0.0192  | 0.6147  |
> > > | MGN + FE      | 0.00817    | 0.01241 | 0.01583 | 0.00721 | 0.60593 |
> > > | Promotion (%) | 30.4       | 41.1    | 45.7    | 62.5    | 1.4     |
> > > | MP-PDE        | 0.0178     | 0.0284  | 0.0386  | 0.0652  | 0.6076  |
> > > | MP-PDE + FE   | 0.01445    | 0.01957 | 0.02621 | 0.03655 | 0.60372 |
> > > | Promotion (%) | 18.9       | 31.2    | 32.1    | 41.5    | 0.6     |
> > > | CeGNN         | 0.0066     | 0.0036  | 0.0024  | 0.0013  | 0.5559  |
> > >
> > > Concretely, the FE block is inspired by interaction models in physics and mathematically hypothesized to capture nonlinear dependencies, enhancing the model's representation power.
> > >
> > > ***Hypothesis.*** The FE block's hypothesis could be framed as: *By capturing second-order interactions between latent features and applying selective filtering, the model can better represent complex structures or relationships in the data*.
> > >
> > > ***Physical Analogy.*** In physics, the outer product and second-order terms are often used to model interactions, such as stress tensors in mechanics or pairwise correlations in quantum mechanics. Here, the module could draw an analogy to systems where interactions between individual components (features) are crucial to the overall behavior.
> > >
> > > ***Process Overview.*** In detail, it regards the node latent feature $\overline{\mathbf{h}}_ i \in\mathbb{R}^{D\times 1}$ as basis and builds a higher-order tensor feature $\mathbf{H}_ {i} \in\mathbb{R}^{D\times D}$ via an outer-product operation, e.g., $\overline{\mathbf{h}}_ {i}\otimes\overline{\mathbf{h}}_ {i}$. This process creates abundant second-order nonlinear terms to enrich the feature map. We then use a mask operation with $\mathbf{M}\in\mathbb{R}^{D\times D}$ to randomly sample these terms, filtering the appropriate information by a learnable weight tensor $\mathbf{W}\in\mathbb{R}^{D\times D\times D}$ to enhance the model's representation capacity.
> > >
> > > In summary, unlike models based solely on topological structures, our work combines both the **cell and FE** modules for efficiently **extracting and processing** features that are essential for learning complex spatiotemporal dependency. This dual-module structure enables our model to better capture dynamic interactions. We have provided substantial theoretical analysis (see **Section 3, Appendix Section C** in the revised paper) and experimental validation (see **Section 4, Appendix Section D**) of our contribution, which differentiates our approach from existing models. We believe that this combination is key to achieving the improvements demonstrated in our experiments and that the theoretical novelty of our work lies in this synergy between topology and feature processing.
> > >
> > > Hope this clarifies your concern.
> > >
> > > ***Concluding Remark:*** We appreciate the reviewer’s additional comments and critical evaluation. We sincerely hope to have your re-evaluation of our paper in light of our clarifications and contributions. Your possible consideration of updating the score is highly appreciated!
> > >
> > > We look forward to your feedback!

---

> > > > ### Comment · Reviewer_1r7X · 2024-11-27
> > > > **Higher order message passing**
> > > >
> > > > Dear Authors,
> > > >
> > > > I understand the importance of second-order interactions in physical modeling. However, there have been works proposing higher-order interaction modeling / message passing for physical systems. Please check:
> > > >
> > > > Predicting molecular properties with covariant compositional networks
> > > > https://pubs.aip.org/aip/jcp/article-abstract/148/24/241745/964420/Predicting-molecular-properties-with-covariant?redirectedFrom=fulltext
> > > >
> > > > Covariant Compositional Networks For Learning Graphs
> > > > https://arxiv.org/abs/1801.02144
> > > >
> > > > Invariant and Equivariant Graph Networks
> > > > https://arxiv.org/abs/1812.09902
> > > >
> > > > Best,
> > > > Reviewer

---

> > > > > ### Author Response · Authors · 2024-11-27
> > > > > **Reply to your comment on high-order message passing**
> > > > >
> > > > > Thank you for listing these papers.
> > > > >
> > > > > Although these methods all involve modeling of high-order interactions, the feature representation of our model is fundamentally different from these existing works. We hope the reviewer will not simply confuse the concepts.
> > > > >
> > > > > Please note that our FE block is uniquely designed, and it is this aspect that sets the novelty of our work. We hope this helps clarify the reviewer’s misunderstanding.
> > > > >
> > > > > Thanks!
> > > > >
> > > > > The Authors

---

> ### Author Response · Authors · 2024-11-25
> **Request your feedback before the end of the discussion period**
>
> Dear Reviewer 1r7X:
>
> As the author-reviewer discussion period will end soon, we would appreciate it if you could review our responses at your earliest convenience. If there are any further questions or comments, we will do our best to address them before the discussion period ends.
>
> Thank you very much for your time and efforts. Looking forward to your response!
>
> Sincerely,
>
> The Authors

---

> ### Author Response · Authors · 2024-11-26
> **Sincerely looking forward to your feedback**
>
> Dear Reviewer 1r7X,
>
> Again, thanks for your constructive comments, which are very much helpful for improving our paper. If there are any further questions or points that need discussion, we will be happy to address them. Your feedback is invaluable in helping us improve our work, and we eagerly await your response.
>
> Moreover, we have thoroughly proofread our paper, corrected typos and grammar mistakes, and re-organized the contents to improve the clarity of the paper. We believe the presentation has been **substantially improved** (see revisions marked in red color). Please refer to the **updated .pdf file**.
>
> Thank you very much for your time and consideration.
>
> Best regards,
>
> The Authors

---

> ### Author Response · Authors · 2024-11-27
> **We eagerly await your response**
>
> Dear Reviewer 1r7X,
>
> As the rebuttal period has undergone over two weeks, your silence made us anxious. We would like to follow up on our rebuttal to ensure that all your concerns have been adequately addressed. If there are any further questions or points that need discussion, we will be happy to address them. We eagerly await your response.
>
> Thank you very much for your time and consideration.
>
> Best regards,
>
> The Authors

---

### Official Review · Reviewer_rgEv · 2024-10-30

**Soundness:** 2
**Presentation:** 3
**Contribution:** 2
**Rating:** 6
**Confidence:** 5

**Summary:**

The authors proposed a cell-embedded GNN model (aka,CeGNN) to learn spatio-temporal dynamics. They claim that their learnable cell attribution to the node-edge message passing process better captures the spatial dependency of regional features.

**Strengths:**

1. The paper is easy to follow.
2. We can see detailed description of the technical details
3. This paper touches the core problem in this area.

**Weaknesses:**

1. The most puzzling aspect of this paper is that the discussion of related work and the selection of baselines are all based on studies from before 2022. In fact, there have been many breakthrough studies on both graph-based methods and other neural operator approaches in recent years [1].

2. This work appears more like a straightforward extension of MP-PDE, both in terms of methodology and experiments. The paper proposes a cell-based method for extracting spatial relationships, but how much improvement could be observed if this feature were integrated into MP-PDE?

3. The main experimental results are somewhat confusing. Since the code is not available, it is unclear whether the training data was generated from the authors' own simulations or from public datasets, and what the training dataset size is. If the data is self-generated, the comparison with a few simple baselines is not convincing. Furthermore, the authors mention long-term simulations, yet all experiments are based on one-step predictions, which is clearly insufficient.

4. Regarding the core innovation of this paper, the cell feature is merely a learnable feature initialized by the distance to the cell center. Can its significance be verified by theoretical analysis or by measuring the distance between cell features and node features? The benefit here might simply be from adding position awareness, which makes the model fit specific data better. It could even be considered to replace the distance to the cell center with the distance to the nearest PDE boundary for each point, which might also yield improvements.

[1] Wang, Haixin, et al. "Recent Advances on Machine Learning for Computational Fluid Dynamics: A Survey." arXiv preprint arXiv:2408.12171 (2024).

**Questions:**

See weakness above

---

> ### Author Response · Authors · 2024-11-21
> **Reply to Reviewer rgEv (Part 1)**
>
> Thanks for your constructive comments!
>
> >**Weakness 1:** Add more recent works as baselines.
>
> **Reply:** Following your suggestion, we have cited several important studies in recent years [1], especially those published after 2022 (see Section 2 on Page 4) and added detailed experimental results of the baseline comparison (see Subsection 4.3 on Page 7, Table 1 on Page 9, Appendix Subsection D.2 on Page 19, Appendix Tables S10-S12 on Pages 20-21) in the revised paper.
>
> Concretely, we have considered some new baselines (e.g., FFNO, Geo-FNO, Transolver) to compare with our CeGNN model on our all cases. A summary of the results of these models is shown in the following **Table A**. In particular, we found that the Geo-FNO [2] and Transolver [4] perform poorly on the limited training dataset, which is significantly smaller, by at least an order of magnitude less, than the hundreds or thousands of datasets typically used for such models. We have added the new results to our revised paper, referring to **Table 1**.
>
> **Table A: RMSE metrics for CeGNN and other models.**
> | Model         | 2D Burgers | 2D FN   | 2D GS   | 3D GS            | 2D BS   |
> | ------------- | ---------- | ------- | ------- | ---------------- | ------- |
> | CeGNN         | 0.00664    | 0.00364 | 0.00248 | 0.00138          | 0.55599 |
> | Geo-FNO [2]    | 0.59363    | 20.514  | 0.18669 | NaN              | 1.2893  |
> | FFNO [3]       | 0.03341    | 0.11921 | 0.03628 | 0.03594 |  --       |
> | Transolver [4] | 0.17422    | 0.13724 | 0.18594 | 0.15204          | 0.81991 |
>
>
> **References:**
>
> [1] Wang, et al. Recent Advances on Machine Learning for Computational Fluid Dynamics: A Survey. arXiv:2408.12171, 2024.
>
> [2] Li, et al. A. Fourier neural operator with learned deformations for pdes on general geometries. arXiv:2207.05209, 2022.
>
> [3] Tran, et al. Factorized fourier neural operators. In ICLR, 2023.
>
> [4] Wu, et al. Transolver: A fast transformer solver for pdes on general geometries. In ICML, 2024 (Spotlight).
>
> >**Weakness 2a:** Is CeGNN an extension of MP-PDE? Whether the cell features enhance the performance of MP-PDE or not.
>
> **Reply:** Thanks for your comment. In fact, our method is **not a straightforward extension of MP-PDE**.
>
> Briefly, the key innovation of MGN-based method, MP-PDE, lies in replacing the MLP in MGN's decoder with a 1D-CNN block to aid autoregressive temporal marching. Furthermore, the training strategy, aka, the **pushforward trick and temporal bundling trick**, also leads to its performance improvement.
>
> However, the innovation of our CeGNN model is the two-level cell-embedded message passing mechanism and the unique feature-enhanced (FE) block. Moreover, all our experiments are trained with an end-to-end **one-step training strategy**, rather than the pushforward trick and temporal bundling trick used in MP-PDE.
>
> Given the cell feature that better captures the spatial dependency and the FE block that further learns the distinguishable feature, CeGNN achieves superior performance compared with other baselines, significantly reducing the prediction errors on all benchmarks, as shown in **Table 1** in the revised paper and **Table A** shown above.
>
> >**Weakness 2b:** How much improvement could be observed if this feature was integrated into MP-PDE?
>
> **Reply:** Following your suggestion, we have conducted experiments incorporating the cell feature in MGN and MP-PDE models and added detailed results in the revised paper (see Subsection 4.3 on Page 7, Appendix Subsubsection D.3.3 on Pages 21-22, Appendix Table S14 on Page 22).
>
> Specifically, we consider MP-PDE integrating the cell feature as our baseline, listed as "MP-PDE + Cell" in the following **Table B**. Meanwhile, the results of MGN integrating the cell feature have been also listed as "MGN + Cell". The results in Table B show that the cell feature improves the performance of MP-PDE, but the "MP-PDE + Cell" still underperforms our model. **More importantly, the results further explain why we integrate the cell feature into MGN rather than MP-PDE.** We have added the new results to our revised paper (referring to **Appendix Table S14**).
>
> **Table B: RMSE metrics for CeGNN, MGN, MGN+Cell, MP-PDE, and MP-PDE + Cell.**
> | Model         | 2D Burgers | 2D FN | 2D GS | 3D GS | 2D BS |
> | ------------- | ---------- | ----- | ----- | ----- | ----- |
> | MGN           | 0.01174    | 0.02108 | 0.02917 | 0.01925 | 0.61475 |
> | MGN + Cell    | 0.00826           | 0.00791        | 0.00832        |   0.00694 | 0.58019 |
> | Promotion (%) | 29.6           |   62.4      |  71.4       |        63.9 |  5.6       |
> | MP-PDE        | 0.01784    | 0.02848 | 0.03860 | 0.06528 | 0.60761 |
> | MP-PDE + Cell | 0.00951           |  0.01193       | 0.00947        |  0.00992       | 0.59313        |
> | Promotion (%) | 46.7          |  58.1       | 75.4        |      84.8   |   2.38      |
> | CeGNN         | 0.00664    | 0.00364 | 0.00248 | 0.00138 | 0.55599 |

---

> ### Author Response · Authors · 2024-11-21
> **Reply to Reviewer rgEv (Part 2)**
>
> >**Weakness 3:** Dataset source; training dataset size; the experiments on the self-generated training data are not convincing.
>
> **Reply:** Thank you for your comments.
>
> >> The source of training data.
>
> In fact, all the data used in this paper are from the **publicly available datasets**. Here, the datasets of four synthetic PDE systems are from the work in [5], while the real-world Black Sea dataset is taken from [6].
>
> **References:**
>
> [5] Rao, et al. Encoding physics to learn reaction–diffusion processes. Nature Machine Intelligence, 2023, 5(7): 765-779.
>
> [6] Lima, et al. Black Sea Physical Reanalysis (CMEMS BS-Currents). Copernicus Monitoring Environment Marine Service (CMEMS), 2020.
>
> >> The training dataset size.
>
> In fact, we have provided the trajectory number for training, validation, and testing, described by the form like (a/b/c) in **Appendix Table S2** (please see Page 18 in the paper). For example, the BS training/validation/testing dataset size is (20/2/2).
>
> >> Experiments on the self-generated training data are not convincing.
>
> In fact, we have experimented not only on the synthetic PDE systems, but also the real-world data, which are from the publically available datasets used in previously published papers. All the results in **Table 1** in the revised paper (Page 9) have shown that CeGNN achieves superior performance compared with other baseline models. Hope this clarifies your concern.
>
> >**Weakness 4:** Long-term simulations with one-step prediction.
>
> **Reply:** Thanks for your comment. All experiment tasks are tested on **long-term prediction** rather than one-step prediction. Once the model is trained (based on one-step prediction), we employ multi-step rollout for long-term prediction. In fact, we have described it in **Subsection 4.2** (see Lines 339-342 on Page 7), shown as follows.
>
> - "*... we mainly focus on **predicting much longer time steps** with lower error and attempt to achieve better generalization ability ... utilize the **one-step training strategy** ...*"
>
> More rigorously, our research task is to predict **all the next $n$ steps** given a random initial condition $\mathbf{u}_ {0}$ taking the autoregressive form $\mathbf{u}_ {0} \rightarrow \mathcal{F}(\mathbf{u}_ {0}) \rightarrow \hat{\mathbf{u}}_ {1} \rightarrow \mathcal{F}(\hat{\mathbf{u}}_ {1}) \rightarrow \hat{\mathbf{u}}_ {2} \rightarrow \cdots \rightarrow \hat{\mathbf{u}}_ {n}$, where the function $\mathcal{F}$ is an unknown function learned by our model, and we set the $n$ steps as **hundreds or thousands of** steps in the synthetic datasets (e.g., 1000 steps in 2D Burgers, 3000 steps in 2D FN, 3000 steps in 2D GS, 3000 steps in 3D GS), and **20** steps in the real-world dataset.
>
> >**Weakness 5:** The significance of cell feature; Can its significance be verified or be measured?
>
> **Reply:** Good question! In fact, the CeGNN's innovation includes the two-level cell-embedded message passing mechanism and the unique feature-enhanced (FE) block. Given the cell-embedded mechanism that better captures the spatial dependency and the FE block that further learns the distinguishable features, CeGNN achieves superior performance compared with other baseline models across all benchmark datasets, as shown in **Table 1** in the revised paper (Page 9).
>
> Following your suggestion, we added explanation in the revised paper (Subsection 3.1 on Pages 4-6 and Appendix Section C on Pages 16-17).
> - A primary theoretical deduction in **Subsection 3.1**.
> - More theoretical preliminaries in **Appendix Section C**.
>
> Generally, the traditional message passing (MP) mechanism can be regarded as a refinement on a discrete space, analogous to an interpolation operation, which implies that edges are essentially interpolated from nodes. A MP mechanism introducing the cell further enhances the refinement of the discrete space (secondary refinement), thereby reducing the magnitude of discretization errors spatially, paving the way for its application in complex graph structures.
>
> ***Definition 1 (Cell in Graph):*** Let $G=(V, E)$ be a graph, where $V$ is the set of nodes $\mathbf{v}$ and $E\subseteq V\times V$ is the set of edges. A cell in $G$ is a subset of nodes $C \subseteq V$, such that the nodes in $C$ form a complete subgraph (clique) or satisfy predefined structural relationships. In particular, a $k$-cell $C_k$ in a graph $G$ contains $k+1$ nodes, where $\forall i, j \in C_k,$, $(\mathbf{v}_i, \mathbf{v}_j) \in E$, representing various structures, such as node ($k=0$), edge ($k=1$), triangle ($k=2$), tetrahedron ($k=3$), and so on.
>
> ***Corollary 1:* (Expressive Power)** Given a graph $G$ including many $k$-cell ($k=0,1,2,\dots$), there exists a cell-based scheme that is more expressive than Weisfeiler-Lehman (WL) tests in distinguishing non-isomorphic graphs (see the proof in **Appendix Subsection C.2** of the revised paper (Pages 16-17)).
>
> Hence, we proposed a new two-level cell-embedded mechanism to process the message on graphs.

---

> ### Author Response · Authors · 2024-11-21
> **Reply to Reviewer rgEv (Part 3)**
>
> >**Weakness 6:** The benefit of adding position awareness; the performance comparison of replacing the distance to the cell center with the distance to the nearest PDE boundary for each point.
>
> **Reply:** Thanks for your comment. Following your suggestion, we tested other two variant models, namely, "CeGNN w/o Cell Pos." and "CeGNN w Cell (Pos. to B)", with the results reported in the following **Table C**. Here, "CeGNN w/o Cell Pos." represents the cell initial feature without the position awareness, and "CeGNN w Cell (Pos. to B)" replaces the distance to the cell center with the distance to the nearest PDE boundary. The results in **Table C** show that the performance improvement of CeGNN is not only due to adding position awareness into the cell initial feature, but also the secondary refinement of the discrete space. We have added the new results to our revised paper (referring to **Appendix Table S15** on Page 22 in the revised paper).
>
> **Table C: RMSE metrics for CeGNN and other two cases.**
> | Model          | 2D Burgers | 2D FN | 2D GS | 3D GS | 2D BS |
> | -------------- | ---------- | ----- | ----- | ----- | ----- |
> | CeGNN (ours)         | **0.00664**    | **0.00364** | **0.00248** | **0.00138** | **0.55599** |
> | CeGNN w/o Cell Pos. |  0.00721          | 0.00477      | 0.00439      |  0.00274     | 0.56098      |
> | CeGNN w Cell (Pos. to B)  | 0.00720 | 0.00490      | 0.00445     |  0.00276     | 0.56040      |
> | MGN           | 0.01174    | 0.02108 | 0.02917 | 0.01925 | 0.61475 |
>
>
> Overall, we appreciate your constructive comments and suggestions. Please let us know if you have any other questions. We look forward to your feedback!

---

> ### Comment · Reviewer_rgEv · 2024-11-22
> **Thank you for your response**
>
> Most of my concern is addressed. But I observe that the time step setting is different between different datasets. Why they have such a large gap. Does it can be attributed to the different time interval setting in different setting? For example, 1000 steps in your 2D Burgers setting takes only a few seconds.

---

> > ### Author Response · Authors · 2024-11-23
> > **Reply to your additional comment**
> >
> > Thank you for your feedback. Your time and effort placed on reviewing our paper are greatly appreciated!
> >
> > > **Question:** Different settings of time interval and time step in the synthetic datasets.
> >
> > **Reply:** Good question! In fact, the numerical accuracy and stability in numerical simulation of physical systems (e.g., 4 synthetic datasets in our work) mainly rely on temporal and spatial discretization (e.g., $\Delta t$, $\Delta x$), the form and coefficients of governing equations, as well as the spefici numerical solver. Typically, when finite-difference-based solvers are employed, the spatiotemporal grids should satisfy a certain condition, e.g., the Courant-Friedrichs-Lewy (CFL) condition.
> >
> > For illustration, the CFL condition in 1D problems is expressed as $\Delta t$ $\leq$ $C\Delta x /v$, where $C$ is a constant related to the specific numerical solver, and $v$ is the maximum propagation speed in the physical process. Given the computational domain and mesh grids, the time interval $\Delta t$ depends on the partial differential equations that govern the specific physical problem. Hence, the time intervals have different settings in different examples.
> >
> > The number of time steps is a parameter we empoyed to test the model's performance of long-term rollout prediction. We choose the corresponding number of time steps when the dynamics get stabilized (e.g., the parterns remain stable), which depends on the specific physical problem and varies in each synthetic dataset.
> >
> > ***Concluding Remark:*** Thank you very much for your constructive comments. Please let us know if you have any other questions. Your consideration of improve the rating of our paper will be much appreciated!

---

> > ### Author Response · Authors · 2024-11-26
> > **Looking forward to your feedback on our reply to your additional comment**
> >
> > Dear Reviewer rgEv,
> >
> > Again, thanks for your constructive comments, which are very much helpful for improving our paper.
> >
> > Moreover, your additional comment on "time interval" has been addressed (please our reply above). If there are any further questions or points that need discussion, we will be happy to address them. Your feedback is invaluable in helping us improve our work, and we eagerly await your response.
> >
> > On a separate note, we have thoroughly proofread our paper, corrected typos and grammar mistakes, and re-organized the contents to improve the clarity of the paper. We believe the presentation has been **substantially improved** (see revisions marked in red color). Please refer to the **updated .pdf file**.
> >
> > Thank you very much for your time and consideration.
> >
> > Best regards,
> >
> > The Authors

---

> > > ### Comment · Reviewer_rgEv · 2024-11-27
> > > **Thanks for your response**
> > >
> > > First, thanks for your response. After reading your rebuttal (including the response to other ), I can understand your attempt to take advantage of the cell-embedded graph model, and I will improve my score correspondingly.
> > >
> > > But considering the comparison effectiveness (baselines are still too old) , novelty of cell-attribution (differences with complex GNNs), and the over-claim for long-term rollout, I think this is still a borderline paper or marginally above the threshold. Hope the authors can provide more insights of how complex representation can help the spatial-temporal modeling. And AC can take a comprehensive decision based on our discussion. Thanks for your effort.

---

> > > > ### Author Response · Authors · 2024-11-27
> > > > **Reply to the additional comments from Reviewer rgEv**
> > > >
> > > > Thank you for your additional feedback. Your time and effort placed on reviewing our paper are greatly appreciated!
> > > >
> > > > > **Comment: Regarding the comparison effectiveness (baselines are "too old").**
> > > >
> > > > **Reply:** We appreciate but respectfully *disagree* with the comment that the baselines used in our study are "too old". In fact, we carefully selected a range of baselines (**eight in total**) that are widely recognized in the field (e.g., MGN [1] (ICLR, 2021), GAT [2] (arXiv, 2017), GATv2 [3] (arXiv, 2022), FNO [4] (ICLR, 2021)) and some latest representative methods (e.g., MP-PDE [5] (ICLR, 2022), FFNO [6] (ICLR, 2023), Geo-FNO [7] (NIPS, 2024), Transolver [8] (ICML, 2024)) for comparison. Meanwhile, we have provided **extensive** experiments (see **Tables 1-5, Appendix Tables S13-S17** in the revised paper) comparing our method with all baselines to demonstrate our model's effectiveness. If the reviewer believes additional specific baselines are necessary, we sincerely welcome your suggestions and would be more than happy to include them in final version of the paper.
> > > >
> > > > > **Comment: Regarding the novelty of cell-attribution (differences from complex GNNs).**
> > > >
> > > > **Reply:** Our method provides a **lightweight yet effective alternative two-level message passing mechanism**, specifically tailored for prediction of spatiotemporal dynamics. Generally, the application of complex spatial GNNs comes with significant training challenge (see **Appendix Table S17** in the revised paper). In contrast, our cell-attribution mechanism introduces rich and effective localized spatial feature learning with efficient two-level message passing mechanism, which is **not only easy to follow but also interpretable** (see **Appendix Subsubsection D.3.5** in the revised paper). The novelty of cell-attribution has been clearly demonstrated from the theoretical perspective (see **Subsubsection 3.1.2, Appendix Subsection C.1, C.2, D.3** in the revised paper) and evidence of the empirical results (see **Tables 1, 3 and 4, Appendix Tables S13-S17** in the revised paper). Hope this clarifies your concern.
> > > >
> > > >
> > > > > **Comment: Regarding the "over-claim for long-term rollout".**
> > > >
> > > > **Reply:** We respectfully clarify that our claims regarding the long-term rollout performance are **fully supported** by extensive experimental results. As shown in **Figures 5-6, Tables 1, 3 and 4, Appendix Tables S13-S17** in the revised paper, our model consistently outperforms the baseline models in long-term prediction tasks across diverse datasets.
> > > >
> > > > As we all know, **long-term rollout is a fundamental and common challenge** in the field of spatiotemporal dynamcis, as error accumulation exists especially when the training datasets are limited. However, our results demonstrate clear improvements of our model over existing methods, which we believe is rational to justify our claim. Hope this clarifies your concern.
> > > >
> > > > > **Comment: Regarding insights into how complex representation helps spatial-temporal modeling.**
> > > >
> > > > **Reply:** We agree that exploring the role of complex representation in spatiotemporal modeling is an important direction. Our study has demonstrated how the two-level message passing mechanism (see **Subsubsection 3.1.2, Appendix Subsection C.1, C.2, D.3, Tables 1-3 and 4, Appendix Tables S13-S17** in the revised paper) and the feature-enhanced block (see **Subsubsection 3.1.1, Appendix Subsection C.3, Tables 2-3** in the revised paper) effectively capture these spatial interactions. In our future work, we plan to further invetigate along this horizon.
> > > >
> > > >
> > > > ***Concluding Remark:*** We appreciate the reviewer's feedback and the suggestion for the AC to take a comprehensive view. We believe our work makes a clear contribution to spatiotemporal prediction by introducing the **two-level message passing mechanism** and the **feature-enhanced block**. We hope our clarifications provided above could thoroughly address the reviewer's concerns. Thank you very much!
> > > >
> > > > ***References:***
> > > >
> > > > [1] Pfaff et al. Learning mesh-based simulation with graph networks. In ICLR, 2021.
> > > >
> > > > [2] Veličković et al. Graph attention networks. arXiv 2017.
> > > >
> > > > [3] Brody S, Alon U, Yahav E. How attentive are graph attention networks? arXiv 2022.
> > > >
> > > > [4] Li et al. Fourier neural operator for parametric partial differential equations. In ICLR, 2021.
> > > >
> > > > [5] Brandstetter et al. Message passing neural PDE solvers. In ICLR, 2022.
> > > >
> > > > [6] Tran et al. Factorized fourier neural operators. In ICLR, 2023.
> > > >
> > > > [7] Li et al. Geometry-informed neural operator for large-scale 3d pdes. In NIPS, 2024.
> > > >
> > > > [8] Wu et al. Transolver: A fast transformer solver for pdes on general geometries. In ICML, 2024.

---

> ### Author Response · Authors · 2024-11-25
> **Request your feedback before the end of the discussion period**
>
> Dear Reviewer rgEv:
>
> As the author-reviewer discussion period will end soon, we would appreciate it if you could review our responses at your earliest convenience. If there are any further questions or comments, we will do our best to address them before the discussion period ends.
>
> Thank you very much for your time and efforts. Looking forward to your response!
>
> Sincerely,
>
> The Authors

---

### Official Review · Reviewer_S14d · 2024-10-30

**Soundness:** 4
**Presentation:** 3
**Contribution:** 4
**Rating:** 10
**Confidence:** 5

**Summary:**

In general, this paper tackles interesting and meaningful problems governed by PDE. It is well written and the results are sound.

**Strengths:**

Ablation study is pretty great to justify the proposed framework.

**Weaknesses:**

The methodology of feature-enhanced is not sufficient, the authors should write down in the appendix more equations with more explanation. Most importantly, why do authors propose such Algorithm 1, is there any physical or mathmatical meaning/inspiration? or any hypothesis? It would be better to show the train of thoughts of how did author propose this FE instead of just showing its working better.

**Questions:**

See weaknesses

---

> ### Author Response · Authors · 2024-11-21
> **Reply to Reviewer S14d**
>
> Thanks for your positive feedback and constructive comments.
>
> >**Weakness:** More theoretical explanation of FE block.
>
> **Reply:** Great suggestion! Following your suggestion, we have added detailed explanation in the revised paper (see Section 3.1.1 on Pages 4-5 and Appendix Section C.3 on Page 17).
>
> - We updated the **FE block in Figure 2** to further explain its process and redefined **Algorithm 1** (see Appendix).
> - A primary theoretical deduction was updated in **Subsubsection 3.1.1**.
> - More theoretical knowledge have also been added in **Appendix Section C.3**.
>
> The FE block is inspired by interaction models in physics and mathematically hypothesized to capture nonlinear dependencies, enhancing the model's representation power.
>
> ***Hypothesis.*** The FE block's hypothesis could be framed as: *By capturing second-order interactions between latent features and applying selective filtering, the model can better represent complex structures or relationships in the data*.
>
> ***Physical Analogy.*** In physics, the outer product and second-order terms are often used to model interactions, such as stress tensors in mechanics or pairwise correlations in quantum mechanics. Here, the module could draw an analogy to systems where interactions between individual components (features) are crucial to the overall behavior.
>
> ***Process Overview.*** In detail, it regards the node latent feature $\overline{\mathbf{h}}_ i \in\mathbb{R}^{D\times 1}$ as basis and builds a higher-order tensor feature $\mathbf{H}_ {i} \in\mathbb{R}^{D\times D}$ via an outer-product operation, e.g., $\overline{\mathbf{h}}_ {i}\otimes\overline{\mathbf{h}}_ {i}$. This process creates abundant second-order nonlinear terms to enrich the feature map. We then use a mask operation with $\mathbf{M}\in\mathbb{R}^{D\times D}$ to randomly sample these terms, filtering the appropriate information by a learnable weight tensor $\mathbf{W}\in\mathbb{R}^{D\times D\times D}$ to enhance the model's representation capacity.
>
> ***Outer Product as Basis Expansion.*** The outer product operation $\otimes$ on the reshaped feature map $\overline{\mathbf{h}}_ {i}\in\mathbb{R}^{D\times 1}$ expands the original latent feature space into a higher-order tensor space. This expansion introduces second-order terms (e.g., $\alpha\beta$ for $\alpha, \beta \in \overline{\mathbf{h}}_{i}$), which can capture interactions between individual components of the original feature $\mathbf{h} _{i} \in\mathbb{R}^{D}$. Mathematically, the second-order tensor reads $\overline{\mathbf{h}} _{i} \otimes\overline{\mathbf{h}} _{i}$.This operation creates a richer feature map with cross-term interactions that may not be explicitly encoded in the original latent space.
>
> ***Lemma 1 (Nonlinear Representation):*** The second-order terms $\alpha\beta$ can model nonlinear dependencies between features. This is particularly useful for capturing complex interactions that linear transformations (e.g., simple dot products) might overlook.
>
> ***Definition 1:*** The FE block expands the latent feature $\overline{\mathbf{h}}_ i \in\mathbb{R}^{D\times 1}$ of node $i$ into a higher-order tensor space using an outer product: $\mathbf{H}_ {i}=\overline{\mathbf{h}}_ {i} \otimes\overline{\mathbf{h}}_ {i}$, where $\mathbf{H}_ {i}\in\mathbb{R}^{D\times D}$ is a higher-order feature map.
>
> ***Regularization via Masking.*** Masking introduces sparsity in $\mathbf{H}_ {i}$, reducing overfitting. If $\mathbf{M}_ {jk}$ is selected, $\mathbf{M}_ {jk} =1$. Otherwise, $\mathbf{M}_ {jk} =0$. Here $j$ and $k$ index the $M \in\mathbb{R}^{D\times D}$ components.
>
> Theoretically, this operation serves two purposes: (1) Reducing computational complexity by randomly sampling terms from the higher-order feature space. (2) Regularizing the model by introducing sparsity, which can prevent overfitting in high-dimensional spaces.
>
> ***Learnable Filtering.*** The learnable weight tensor $\mathbf{W}\in\mathbb{R}^{D\times D\times D}$ acts as a filter, selecting and emphasizing the most informative terms.
>
> ***Definition 2:*** A mask operation $\mathbf{M} \in \mathbb{R}^{D\times D}$ is applied to randomly sample elements in $\mathbf{H}_ {i}$, and the resulting masked tensor is processed using a learnable weight tensor $\mathbf{W} \in \mathbb{R}^{D\times D\times D}$ as follows: $\tilde{\mathbf{h}}_ i$ = ($\mathbf{M} \odot\mathbf{H}_ {i}): \mathbf{W}$, where $\odot$ represents element-wise multiplication and $:$ denotes double contraction of tensors. The resulting feature $\tilde{\mathbf{h}}_ {i} \in\mathbb{R}^{1\times D}$ enriches the representation of $\mathbf{h}_ {i} \in \mathbb{R}^{D}$.
>
> ***Corollary 1 (Representation Power):*** The full feature map $\mathbf{H}_i$ contains $D^2$ terms for a $D$-dimensional input feature vector $\mathbf{h}_i$. After masking, the effective representation space reduces by the sparsity of $\mathbf{M}$. The learnable filter $\mathbf{W}$ further narrows this down to the most critical terms.
>
> Thank you very much!

---

> > ### Comment · Reviewer_S14d · 2024-11-21
> > **Thank you!**
> >
> > The authors have addressed my comments.

---

> > > ### Author Response · Authors · 2024-11-21
> > > **Thank you for your positive feedback**
> > >
> > > Dear Reviewer S14d,
> > >
> > > Thank you very much for your positive feedback. Your time and effort placed on reviewing our paper are highly appreciated!
> > >
> > > Best regards,
> > >
> > > The Authors

---

### Author Response · Authors · 2024-11-21
**General reply**

Dear Reviewers:

We would like to thank you for your constructive comments, which are very helpful in improving our paper. We are pleased that the reviewers recognized the novelty and effectiveness of our work. In particular, we thank the reviewers for recognizing the *novelty* (S14d, rgEv, and KT6u) and *effectiveness* (S14d, 1r7X, and KT6u) of our method.

We have addressed all the concerns in each individual rebuttal and summarized as follows. Comprehensive revisions and adjustments (indicated in red color) have also been made in the revised paper (please see the **updated .pdf file**).

- We updated the **FE block in Figure 2** to further explain its process and redefined **Algorithm 1** (placed in Appendix).
- We added some new related works and discussed them in **Section 2**.
- We added some results of new baselines on all benchmarks in **Table 1** and a discussion in **Subsection 4.3**.
- More details of new baselines were added in **Appendix Subsection D.2, Table S3, S10-S12**.
- A primary theoretical deduction of the FE block was updated in **Subsubsection 3.1.1**.
- More theoretical analysises of the FE block were added in **Appendix Subsection C.3**.
- A primary theoretical deduction of the CellMPNN block was updated in **Subsubsection 3.1.2**.
- More theoretical analysises of the CellMPNN block were added in **Appendix Subsection C.1-C.2**.
- More detailed results and discussions were added in **Appendix Subsubsection D.3.3-D.3.5, Appendix Table S14-S17**.

Please do feel free to let us know if you have any further questions.

Thank you very much.

Best regards,

The Authors of the Paper

---

> ### Comment · Reviewer_S14d · 2024-11-26
> **The authors deserve some feedback for their effort.**
>
> This is an excellent paper, particularly after addressing all the reviewers' comments. It is well-written, well-organized, and reflects a high level of effort and attention to detail. I firmly believe the authors deserve constructive communication or feedback to acknowledge their dedication.

---

> > ### Author Response · Authors · 2024-11-27
> > **Thank you for your encouraging feedback**
> >
> > Dear Reviewer S14d,
> >
> > Thank you for your kind and encouraging feedback. We very much appreciate your recognition of our contribution!
> >
> > We are sincerely looking forward to the discussion with the reviewers, whose comments have been helpful in improving our paper.
> >
> > Best regards,
> >
> > The Authors

---

### Meta-Review · Area_Chair_6m7X · 2024-12-23

**Metareview:**

The paper proposes an encode-process-decode framework leveraging graph neural networks (GNNs) to model the spatio-temporal dynamics of partial differential equations (PDEs). It extends the classical message-passing paradigm in GNNs by introducing two key components: (i) leveraging higher-order features between graph nodes to capture spatial dependencies beyond immediate neighbor edges, and (ii) enriching node features through higher-order tensor representations to maintain feature distinction and mitigate the over-smoothing issue caused by successive feature aggregations during message-passing operations. The model is evaluated against baseline models across a variety of datasets and demonstrates improved performance.

The reviewers acknowledge the experimental improvements over the baselines. However, they raised several concerns about the initial version of the manuscript. These include the incremental nature of the technical contribution—several reviewers pointed out the existence of alternative methods for capturing higher-order features in GNNs—and the choice of baselines for comparison.
In response, the authors provided extensive new comparisons during the rebuttal phase, including implementing the proposed components on other backbones, which also demonstrated improved performance. While this significantly improved upon the first version of the paper, the majority of the reviewers maintained their initial concerns, particularly regarding the incremental value of the contribution.

**Additional Comments On Reviewer Discussion:**

The authors provided extensive responses to the reviewers' concerns, including new experiments. However, this did not alleviate the reviewers' concerns about the incremental nature of the technical contribution.

---

### Decision · Program_Chairs · 2025-01-22

Reject